



# Ice thickness and bed topography of Jostedalsbreen ice cap, Norway

Mette K. Gillespie[1], Liss M. Andreassen[2], Matthias Huss[3,4,5], Simon de Villiers[1], Kamilla H. Sjursen[1], Jostein Aasen[2], Jostein Bakke[6], Jan M. Cederstrøm[6], Hallgeir Elvehøy[2], Bjarne Kjøllmoen[2], Even Loe[7], Marte Meland[8], Kjetil Melvold[2], Sigurd D. Nerhus[1], Torgeir O. Røthe[1], Eivind W. N. Støren[6,9], Kåre Øst[10], Jacob C. Yde[1]

[1]Department of Civil Engineering and Environmental Sciences, Western Norway University of Applied Sciences, Sogndal, 6851, Norway
[2]Section for Glaciers, Ice and Snow, Norwegian Water Resources and Energy Directorate (NVE), Oslo, 0301, Norway
[3]Laboratory of Hydraulics, Hydrology and Glaciology (VAW), ETH Zurich, Zurich, 8092, Switzerland
[4]Swiss Federal Institute for Forest, Snow and Landscape Research (WSL), Birmensdorf, CH-8903, Switzerland
[5]Department of Geosciences, University of Fribourg, Fribourg, 1700, Switzerland
[6]Department of Earth Science and the Bjerknes Centre for Climate Research, University of Bergen, 5020, Norway
[7]Statkraft, Gaupne, 6868, Norway
[8]Breheimsenteret, Jostedalen, 6871, Norway
[9]COWI, Bergen, 5824, Norway
[10]Norgesguidene, Jostedalen, 6871, Norway

*Correspondence to*: Mette Kusk Gillespie (mette.kusk.gillespie@hvl.no)

**Abstract.** We present an extensive dataset of ice thickness measurements from Jostedalsbreen ice cap, mainland Europe's largest glacier. The dataset consists of more than 351 000 point values of ice thickness distributed along ~1100 km profile segments that cover most of the ice cap. Ice thickness was measured during field campaigns in 2018, 2021, 2022, and 2023 using various ground-penetrating radar (GPR) systems with frequencies ranging between 2.5 and 500 MHz. The large majority of ice thickness observations were collected in spring using either snowmobiles (90 %) or a helicopter-based radar system (8 %), while summer measurements were carried out on foot (2 %). To ensure accessibility and ease of use, metadata were attributed following the GlaThiDa dataset and follows the FAIR (Findable, Accessible, Interoperable, and Reusable) guiding principles. Our findings show that glacier ice of more than 400 m thickness is found in the upper regions of large outlet glaciers, with a maximum ice thickness of ~630 m in the Tunsbergdalsbreen outlet glacier accumulation area. Thin ice of less than 50 m covers narrow regions joining the central part of Jostedalsbreen with its northern and southern parts, making the ice cap vulnerable to break-up with future climate warming. Using the point values of ice thickness as input to an ice thickness model, we compute 10 m grids of ice thickness and bed topography that cover the entire ice cap. From these distributed datasets we find that Jostedalsbreen has a mean ice thickness of 154 m ±22 m and a present (~2020) ice volume of 70.6 ±10.2 km$^3$. Locations of depressions in the map of bed topography are used to delimitate the locations of potential future lakes, consequently providing a glimpse of the landscape if the entire



Jostedalsbreen melts away. Together, the comprehensive ice thickness point values and ice cap-wide grids serve as a baseline for future climate change impact studies at Jostedalsbreen.

All data are available for download at https://doi.org/10.58059/yhwr-rx55 (Gillespie et al., 2024).

## 1 Introduction

Global glacier mass loss caused by increased atmospheric temperatures contributes significantly to changes in sea level, water resources and natural hazards (IPCC, 2021). Projections of future changes show that glaciers and ice caps will continue to lose mass due to anthropogenic warming, and that the majority of the world's glaciers and ice caps are at risk of being lost by 2100 (Rounce et al., 2023). However, global glacier projections remain uncertain. This is especially true for ice caps, where model efforts of ice thickness distribution in the flat upper

regions and across ice divides represents a particular challenge (Millan et al., 2022; Frank et al., 2023).

Information on ice thickness distribution of a glacier is a prerequisite for accurate modelling of ice dynamics and glacier evolution, as well as future hydrological impacts. Ice thickness measurements are also essential for precise calculations of the ice volume of glaciers and in mapping of the subglacial topography. Consequently, significant

efforts have been made to compile ice thickness data and provide grids of ice thickness and bed topography (e.g., Gärtner-Roer et al., 2014; Lindbäck et al., 2018; Frémand et al., 2023). The third version of the Glacier Thickness Database (GlaThiDa v3) includes nearly 4 million ice thickness measurements distributed over roughly 3000 glaciers worldwide, and 14 % of the world's glacierized area is now within 1 km of an ice thickness measurement (GlaThiDa Consortium, 2020; Welty et al., 2020). Direct inter- and extrapolation of ice thickness measurements

with various techniques, such as kriging, inverse-distance weighting, or spline interpolations (Flowers and Clarke, 1999; Binder et al., 2009; Fischer, 2009; Yde et al., 2014; Andreassen et al., 2015) is possible, but may produce large uncertainties in areas without measurements (Gillespie et al., 2023). Consequently, ice thickness modelling is necessary to extrapolate measurements more accurately to unmeasured regions (Andreassen et al., 2015; Farinotti et al., 2021), and to infer ice thickness for glaciers without direct measurements.


Various ice thickness inversion approaches exist that do not require bed topography or ice thickness as input (e.g., Huss and Farinotti et al., 2012; Linsbauer et al., 2012; Fürst et al., 2017; Farinotti et al., 2019; Frank et al., 2023), and recent efforts to model ice thickness through inversion of surface topography have made distributed ice thickness information available for every individual glacier in the world (Farinotti et al, 2019; Millan et al., 2022) and

all Scandinavian glaciers and ice caps (Frank and van Pelt, 2024). Although ice thickness observations are not required as input in these models, databases of ice thickness, when available, remain important for calibration and validation of model behaviour. Assessments of model performances, such as the first Ice Thickness Model



Intercomparison eXperiment (ITMIX; Farinotti et al., 2017), found that model output is highly variable, and that the best results are achieved when using model ensembles. In addition, a more recent model comparison (ITMIX2; Farinotti et al., 2021) demonstrated the added value of in situ ice thickness observations to constrain models. A limited set of ice thickness observations, preferably from the thickest parts of the glacier, provided efficient in constraining mean glacier thickness, illustrating that even sparse ice thickness observations are of importance in ice thickness modelling. Consequently, readily accessible ice thickness observations for calibration and validation remains key for developing a new generation of ice thickness estimation models (Farinotti et al., 2017). Measurements across the flat upper regions of ice caps such as Jostedalsbreen are of particular value, as these can be applied to improve ice thickness models for the much larger ice sheets in Greenland and Antarctica, and ultimately facilitate more accurate predictions of future sea-level change (Morlighem et al., 2017).

In Norway, numerous field campaigns to measure ice thickness have been carried out over the years (Andreassen et al., 2015). The purpose of the earliest measurements was typically to determine subglacial topography in relation to hydropower planning, such as subglacial intakes and water divides (e.g., Kennett, 1989; 1990), or detailed studies related to jökulhlaups (Engeset et al., 2005). While the first attempts at ice thickness mapping used seismic measurements (e.g., Sellevold and Kloster, 1964) or hot water drilling (e.g., Østrem et al., 1976), from 1980 ground-penetrating radar (GPR) has been the preferred method for largescale mapping of glaciers in Norway (e.g., Sætrang and Wold, 1986). Since these first radar measurements on Norwegian glaciers, technological advancements in radar systems, processing techniques and positioning accuracy have enabled the use of GPR in a wide range of glaciological applications, such as mapping of ice- or snow thickness, internal layering, thermal regime, or englacial meltwater channels (e.g., Plewes and Hubbard, 2001; Dowdeswell and Evans, 2004; Navarro and Eisen, 2009). The penetration depth and level of detail in GPR data are determined by the antenna frequency. Information on ice and snow characteristics can be achieved by using very-high (30–300 MHz) or ultra-high (300–3000 MHz) antenna frequencies, while high-frequency GPR surveys (3–30 MHz antenna frequency) have larger penetration depth at the expense of resolution (Schlegel et al., 2022). High-frequency antennas are consequently the better choice in surveys of bed topography and grids of glacier geometry based on such measurements have been widely used to model future changes in Norwegian glaciers (e.g., Laumann and Nesje, 2009, 2014; Giesen et al., 2010; Åkesson et al., 2017, Johansson et al., 2022).

Jostedalsbreen is the largest ice cap in mainland Europe and makes up about 20 % of the total glacierized area of mainland Norway (Andreassen et al., 2022). The effect of global warming is evident in the region and monitored outlet glaciers flowing from the ice cap have thinned and retreated with increased speed since 2000 (e.g., Andreassen et al., 2020; Seier et al., 2024). The effects of future warming on accessibility, glacier-atmosphere systems and hydrology are likely to significantly impact regional businesses such as agriculture, tourism, and





hydropower production. Despite the importance of Jostedalsbreen to both regional stakeholders and the scientific community, the natural and societal consequences of climate-forced changes in the region remain largely unknown.
Future changes of Jostedalsbreen can be assessed through glacier evolution modelling, but accurate results
require high-quality information on ice thickness and bed topography as model input (Farinotti et al., 2017). Although several surveys of ice thickness were conducted on Jostedalsbreen during the 1970s and 1980s (e.g., Østrem et al., 1976; Andreassen et al., 2015), prior to the new ice thickness measurements described in this paper, many parts of the ice cap had either poor or no data coverage.

Here we present a comprehensive and up-to-date point dataset of ice thicknesses of Jostedalsbreen measured by GPR during the period 2018–2023. Ice thickness measurements were predominantly performed on the glacier surface (ground-based), but in regions that were inaccessible on the ground we applied a helicopter (airborne) radar system. We used antenna frequencies ranging from 2.5 to 500 MHz to capture the thickness of the ice in the best possible resolution. For regions that remain unmeasured due to resource or accessibility constraints, we use
interpolation and ice thickness modelling to provide new grids of ice thickness and bed topography for the entire ice cap. Depressions in the subglacial bed topography grid are used to infer the locations of lakes if Jostedalsbreen disappeared completely from the landscape. We provide a thorough description of the uncertainties associated with ice thickness measurements and modelling results, including comprehensive uncertainty estimates. The enhanced datasets on Jostedalsbreen ice thickness and bed topography have the potential to significantly advance
modelling efforts for the past and future evolution of the ice cap and provide accurate assessments of regional climate change impact. In addition, comprehensive high-accuracy measurements over the complex glacier geometry at Jostedalsbreen constitute a valuable resource for improving current ice thickness models, particularly on ice caps, where the flat upper regions and discontinuities across ice divides provide a special challenge.

**2 Study site**

Jostedalsbreen (Fig. 1) has an area of 458 km² and an elevation ranging between 380 and 2006 m a.s.l. (Andreassen et al., 2022). The climate is subarctic to tundra with a mean annual air temperature of -3°C at 1633 m a.s.l. (2009–2022 average at Steinmannen meteorological station; Engen et al., in review). In the most recent national glacier inventory, Jostedalsbreen is divided into 81 glacier units from observations of topographic ice divides (Andreassen et al., 2022). Many of these glacier units have individual names which will be referred to
throughout this paper. Jostedalsbreen is defined as a single ice cap but can geographically be divided into three minor ice caps that are currently connected (Fig. 1). In this paper, we refer to Jostedalsbreen South (south of Grensevarden), Central (north of Grensevarden as far as and including Lodalsbreen glacier) and North (northeast of Lodalsbreen glacier).



Jostedalsbreen reached its maximum Little Ice Age (LIA) extent between 1740 and 1860 CE with an estimated area of 572 km² (Carrivick et al., 2022; Andreassen et al., 2023). Since then, the ice cap has experienced an overall reduction in size, interrupted temporarily by advances in several fast-responding outlet glaciers, the latest of which occurred in the 1990s due to increased winter precipitation (Nesje et al., 1995; Andreassen et al., 2005). By 2006, the major outlet glaciers had in combination lost at least 93 km² or 16 % of their LIA area and 14 km³ or 18 % of

their LIA volume (Carrivick et al., 2022). Increasing summer temperatures further reduced the glacier area by 3 % from 2006 to 2019 (Andreassen et al., 2022) and continues to this day (Seier et al. 2024). Overall, the change in the glacial landscape has been considerable, with measurements of glacier front variation (length changes) at several outlet glaciers revealing a total reduction in length of 1–3 km since ~1900 (Andreassen et al., 2023), of which 300–700 m has occurred since 2000 (Kjøllmoen et al., in prep.).


The first ice thickness measurements on Jostedalsbreen were conducted in 1973 along two cross profiles located between 700 and 800 m a.s.l. on the tongue of Nigardsbreen outlet glacier (Østrem et al., 1976). In total, 14 points were drilled using electrical hot-point drilling, revealing ice thicknesses of up to 200 m. In 1986 hot water drilling was carried out on Bødalsbreen outlet glacier along three cross profiles at 780–815 m a.s.l. (Haakensen and Wold,

1986). Results from 15 boreholes show that ice thickness varied between 50 and 60 m in this region. GPR was first used on Jostedalsbreen in the 1980s during field campaigns on Nigardsbreen and surrounding glaciers in 1981, 1984, and 1985 (Sætrang and Wold, 1986), on Austdalsbreen and surrounding glaciers in 1986 (Sætrang and Holmqvist, 1987), and south of Nigardsbreen in 1989 (Andreassen et al., 2015). Results show that ice thickness along transects typically varied between 150 and 300 m, with ice of up to 600 m in the flattest regions and thinner

ice (50–100 m) at the highest points of the ice cap (Sætrang and Wold, 1986). These early measurements of ice thickness are associated with relatively large uncertainties in surface elevations and the positioning of GPR profiles. In addition, as data were collected and processed with analogue techniques, only parts of the older dataset are available digitally. Digitised data from these campaigns have been submitted to the GlaThiDa database (GlaThiDa Consortium, 2020; Welty et al., 2020) and were used by Andreassen et al. (2015) to interpolate ice thickness

distribution and estimate a mean ice thickness of 158 m for parts of Jostedalsbreen (65 % of total area). More recently, Jostedalsbreen was included in a modelling study of ice volume and thickness distribution of all Scandinavian glaciers (Frank and van Pelt, 2024). In this study, existing ice thickness measurements were used to calibrate an ice thicknesses model, resulting in a total volume of 72.6 km³ for Jostedalsbreen.



## 3 Methods and data

### 3.1 Ice thickness measurements

The ice thickness measurements presented in this paper were collected during field campaigns between 2018 and 2023. The first measurements were carried out in April 2018, however most of the data were gathered in April 2021, March to April 2022 and April 2023 (Fig. A1), while the tongue of Austerdalsbreen was surveyed in September 2021. The principle means of transport during data collection was snowmobile (90 % of all datapoints), but a newly developed helicopter radar system (Air-IPR) was deployed in steep and crevassed regions of the ice cap (8 % of all datapoints). Summer measurements on foot account for only 2 % of all datapoints (Fig. 2). Although airborne surveys were quicker, ground-based measurements were preferred whenever possible due to the generally better data quality caused by lower travel speeds, less noise (electronic and off nadir-reflections) and simpler wave propagation (lack of an air layer). Depending on the surface conditions, we collected the data in a grid pattern, with the main profiles spaced no more than 400 m apart and oriented transverse to the ice flow direction. Survey lines perpendicular to main profiles were 400–800 m apart, depending on accessibility and time constrains during the fieldwork. In total, we have successfully detected the glacier bed along ~920 km of profile segments collected with the ground-based radar systems and ~170 km of profile segments collected with the airborne radar system (Fig. 1). Following the new measurements, 90 % of the ice cap is now less than 300 m from an observation of ice thickness (measurement or glacier outline) and 49 % is within 100 m of a known point.



**Figure 1: Map showing (a) the location of Jostedalsbreen in southern Norway, (b) Jostedalsbreen and GPR surveys divided into helicopter, snowmobile, and foot, and (c) the measurements on Austerdalsbreen by foot and helicopter. The shown glacier extent and outline of glacier units are from 2019 (Andreassen et al., 2022). Background mountain shadow on (c) is from the 100 m national DTM by the Norwegian Mapping Authority. The coordinate systems are geographical coordinates on (a) and UTM 33N, datum ETRS89 on (b) and (c).**

Based on the terminology proposed by Schlegel et al. (2023), we used a combination of high, very high and ultra-high frequency radar systems to gather detailed information on snow, firn and shallow ice, while maintaining a good penetration depth for deep ice. Usually two snowmobiles would travel together, one towing a high frequency generation 1–3 Blue System Integration Ltd. IceRadar system with 2.5 or 5 MHz antennas (Mingo and Flowers, 2010) depending on the ice thickness in the investigated area, and the other snowmobile towing either a higher frequency Malå GPR system with 25 or 50 MHz rough terrain antennas, or 450 or 500 MHz shielded antennas (Table 1). On one occasion, measurements were conducted using a Radarteam GPR system with a 40 MHz monostatic antenna and an upgraded non-commercial GPR with 5 MHz antennas (NVE-radar), similar to that




described by Sverrisson et al. (1980) and Pettersson et al. (2011). For the measurements on foot on the tongue of Austerdalsbreen, we chose a 10 MHz Blue System Integration Ltd. IceRadar and a 50 MHz Malå GPR. All helicopter measurements were collected using a 5 MHz Air-IPR Generation 3 Blue System Integration Ltd.

IceRadar system with the antennas in a V dipole configuration (Table 1). The carrying platform for the Air-IPR is built with wood and uses telescopic rods in composite material to hold the antennas (Fig. 2c). To secure an accurate distance between the antennas and the ice surface, we used a laser mounted on the platform with a wireless connection to the cockpit. The control of the IceRadar during both ground-based and airborne measurements was performed using a tablet and a remote connection.


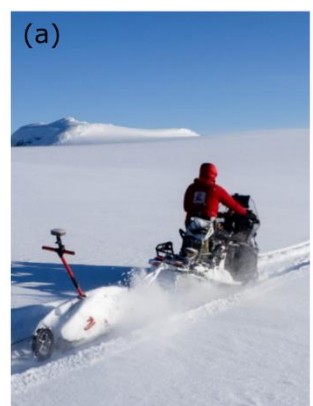 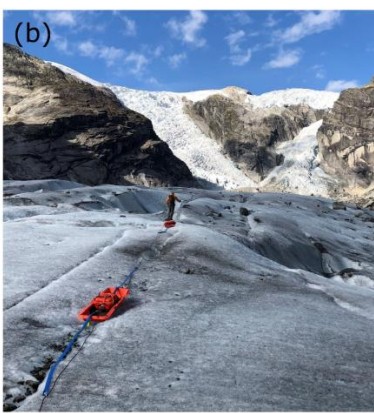 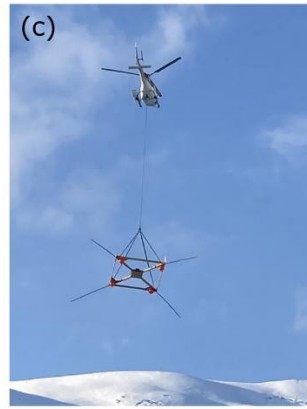

**Figure 2: Data collection was undertaken (a) by snowmobile, (b) on foot, and (c) by helicopter. Photos: (a) Kjetil Melvold, (b) Mette K. Gillespie and (c) Torgeir O. Røthe.**

Ground-based measurements of ice thickness were largely carried out using an in-line antenna configuration with distances between receiver (Rx) and transmitter (Tx) units depending on the antenna frequency, and varying from 4 m (50 MHz) and 6.5 m (25 MHz) for the two Malå rough terrain antennas to 15 m (10 MHz), 30 m (5 MHz) and 60 m (2.5 MHz) for the three IceRadar antenna sets. The 5 MHz NVE-radar antennas were also run using an in-line configuration, but with 32 m between antenna mid-points. By contrast, the shielded 450 MHz and 500 MHz

Malå antennas were oriented perpendicular to the travel direction and with a 0.18 m antenna separation. To avoid interference between radar systems during data collection, the two snowmobiles travelled at a distance of more than 50 m. For frequencies of 25 MHz and above, each measurement (trace) was stacked between 4 and 8 times to increase the signal-to-noise ratio, whereas the 2.5 and 5 MHz measurements were stacked 256 times. Ice thickness measurements were collected at a constant time interval, which varied according to limitations in the

different radar systems. The distance between individual traces along radar profiles was affected by this and our travel speed (~15 km h$^{-1}$). Measurements collected with antenna frequencies ranging between 25 and 500 MHz



were sampled at the highest rate (trace distances of ~0.2–2 m). Therefore, while these measurements constitute a significant proportion of total datapoints (Table 1), the vast majority of data coverage is attributed to ice thickness observations along 5 and 2.5 MHz profiles, which were collected less densely. In general, ground-based

measurements of ice thickness were registered at intervals ranging between 3 and 6 m, while airborne measurements were 3 to 20 m apart.

GNSS locations along survey lines were recorded every 1 s with a horizontal positioning accuracy of up to 5 m for the Malå radar system (G-Star IV BU-353S4 receiver) and 3 m for the IceRadar system (Garmin GPSx OEM

sensor). In addition, differential GNNS (DGNSS) measurements were carried out independently of the radar measurements in some regions.

**Table 1: Survey dates and equipment used for ice thickness measurements during the 2018–2023 field campaigns. The number of datapoints refers to the post-processed and interpreted dataset. Institutions are Western Norway University**
**of Applied Sciences (HVL), the Norwegian Water Resources and Energy Directorate (NVE) and University of Bergen (UIB).**

| Method | Radar type | Frequency | Points | Survey dates | Institutions |
|---|---|---|---|---|---|
| *Ground-based radar* | IceRadar | 2.5 MHz | 15712 | 18–19 April 2018 | HVL |
| | NVE-radar | 5 MHz | 18569 | 18 April 2018 | NVE |
| | IceRadar Malå GPR Malå GPR | 2.5 and 5 MHz 50 MHz RTA 450 MHz shielded | 99745 4503 15308 | 11–18 April 2021 | HVL |
| | RadarTeam Subecho 40 | 40 MHz | 32533 | 16–17 April 2021 | NVE |
| | IceRadar Malå GPR | 2.5 MHz 25 MHz RTA | 5221 5753 | 20–24 April 2021 | UIB |
| | IceRadar Malå GPR | 10 MHz 50 MHz RTA | 4825 2723 | 4 September 2021 | HVL |
| | IceRadar | 5 MHz | 11769 | 8 March 2022 | HVL |
| | IceRadar Malå GPR | 5 MHz 25 and 50 MHz RTA | 18424 11938 | 19–22 March 2022 | HVL |
| | IceRadar | 5 MHz | 5856 | 5–6 April 2022 | NVE |
| | IceRadar Malå GPR Malå GPR | 5 MHz 50 MHz RTA 500 MHz shielded | 53061 12509 4282 | 20–21 April 2022 | HVL |
| | IceRadar | 2.5 MHz | 621 | 22 March 2023 | HVL |
| *Airborne radar* | IceRadar | 5 MHz | 5725 | 22 March 2022 | UIB |
| | IceRadar | 5 MHz | 5151 | 7 April 2022 | UIB and HVL |
| | IceRadar | 5 MHz | 5267 | 26 April 2022 | HVL |
| | IceRadar | 5 MHz | 12064 | 20 April 2023 | HVL |



## 3.2 Data processing and interpretation

The raw GPR data was primarily processed using the ReflexW module for 2D data analysis (Sandmeier Scientific
Software, version 8.5). Initial data processing involved adding GNSS positions for antenna midpoints to all traces,
merging individual shorter profiles into larger segments, and assigning a constant trace increment along each
segment to allow for subsequent migration. We chose a trace increment close to the mean value during travel to
avoid deleting or introducing too many traces to the original dataset. Following the initial data sorting, we used a
combination of 1) dewow, 2) Butterworth bandpass filtering, 3) time zero correction, 4) dynamic correction, 5)
energy decay gain, and 6) f-k Stolt migration on all ground-based measurements. For the GPR measurements
collected with 2.5 and 5 MHz systems, processing steps 3) and 4) are important to account for the influence of the
large antenna separation on first signal arrival times and the radar wave path through the ice. Further filtering was
required on the airborne measurements due to significant system-related noise. The processing routine for this
portion of the dataset consequently involved applying an adaptive filter using the IceRadarAnalyzer processing
software (Blue System Integration Ltd., version 6.3.1. beta) to remove unwanted signals from the radar profiles, in
addition to dewow and bandpass filtering. Subsequent static correction was undertaken in ReflexW using manually
delineated arrival times of the glacier surface reflection, after which energy decay gain and f-k Stolt migration were
applied.

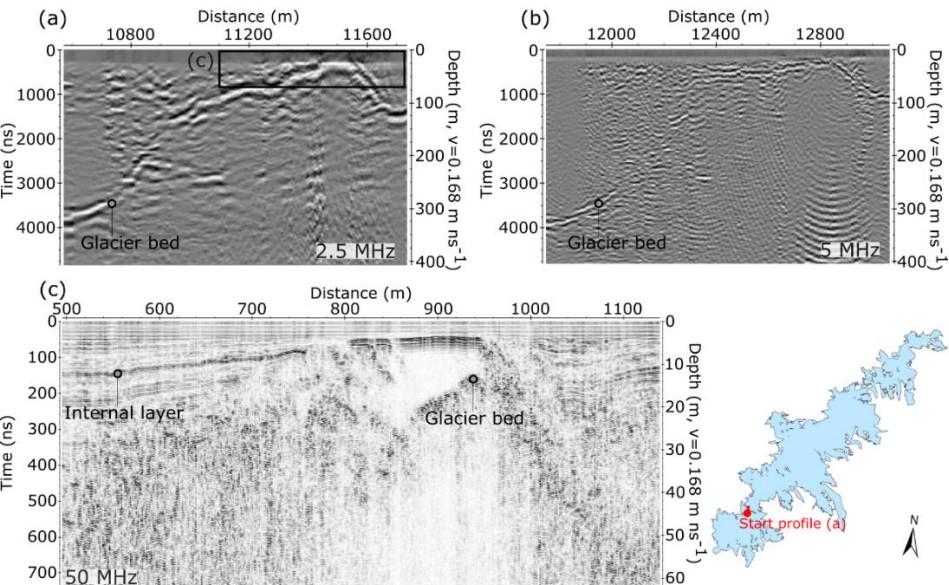

**Figure 3: Example of measurements with (a) 2.5 MHz, (b) 5 MHz and (c) 50 MHz antennas on shallow ice along a profile
travelling north near Grensevarden (Fig. 1). The 2.5 and 50 MHz profiles were collected along identical tracks in 2021,
while the 5 MHz measurement are from 2022 along a profile located ~50 m from these tracks. The radargrams illustrate
well the difference in resolution and penetration depth resulting from variations in antenna frequency. The lowest
frequency measurements provide information on bed topography along the entire profile, while the 50 MHz profile
allows for accurate measurements of thin ice and offers evidence of internal ice characteristics.**



Following data processing, we observed a bed reflection along most 2.5 and 5 MHz radar segments and in higher frequency measurements collected in ice-marginal regions (Fig. 3). The bed reflections were delineated manually, and we calculated ice thickness from the reflection two-way travel time by assuming a constant radio-wave velocity

in ice of 0.168 m ns$^{-1}$, similar to that used on other glaciers in Norway and abroad (Dowdeswell and Evans, 2004; Navarro and Eisen, 2009; Andreassen et al., 2012a; Yde et al., 2014; Johansson et al., 2022). The range of frequencies allows for a detailed mapping of both shallow and deep ice at the best possible resolution. In shallow regions, ice thickness was most accurately determined from the highest frequency measurements, which also provide information on snow (450 and 500 MHz data only), firn and internal layer characteristics (Fig. 3c). In this

paper, we present only the interpreted ice thickness from these higher frequency measurements. In general, GPR measurements at Jostedalsbreen are characterised by strong scattering and rapid attenuation of the radar signal (Fig. 3c), as is typical for radar surveys on temperate glaciers (Smith and Evans, 1972; Ogier et al., 2023). Occasionally, regions of more transparent ice were observed in the higher frequency measurements (Fig. 3c). These likely indicate either zones that are above the internal water table or isolated patches of cold (frozen) ice.

While the 5 MHz antennas generally performed well in depths of up to 400–500 m, the advantage of using 2.5 MHz antennas was evident in areas with sloping bed topography (Fig. 3b) and in the deepest regions, where reflectors were sometimes weak or absent, even with the 2.5 MHz system (Fig. 4).

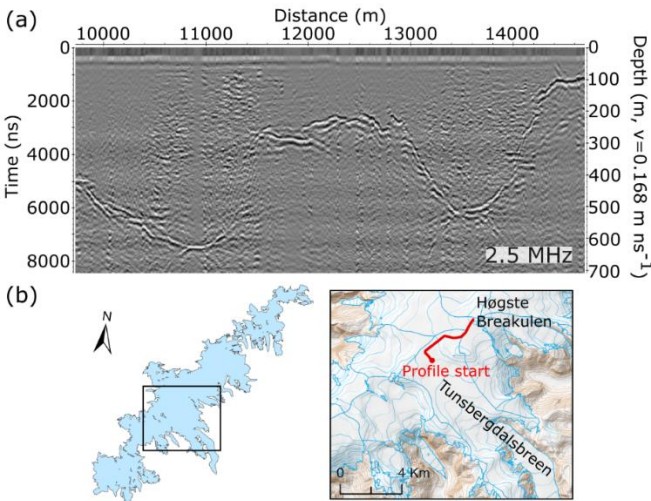

**Figure 4: (a) Example radargram of measurements with 2.5 MHz antennas. (b) The profile was located along a transect in the upper part of Tunsbergdalsbreen (Fig. 1), where the thickest ice was observed. The detailed background map in (b) is from the Norwegian Mapping Authority (WMS for Topografisk Norgeskart available at https://www.geonorge.no/) and the 2019 outline of glacier units on (b) is from Andreassen et al. (2022).**





The efficiency of snowmobile transport during the fieldwork depended strongly on the snow conditions and varied significantly between field seasons. For example, valley access onto Tunsbergdalsbreen was possible in 2022, when the snow cover was thick, but attempts to drive onto the glacier tongue in 2023 had to be abandoned. The helicopter measurements generally cover regions that were inaccessible on snowmobile, either due to steep and/or crevassed terrain, or unfavourable snow conditions. Consequently, helicopter measurements provide a valuable

addition to the ground-based measurements. However, the airborne measurements generally had a lower penetration depth than ground-based measurements using the same antenna frequency, primarily due to increased electronic noise and radar wave attenuation, as well as scattering of the radar signal caused by large surface crevasses present in many airborne surveyed regions. Despite these challenges, bed reflectors were generally observed at depths of up to 350–400 m of ice in airborne measurements (Fig. 5).


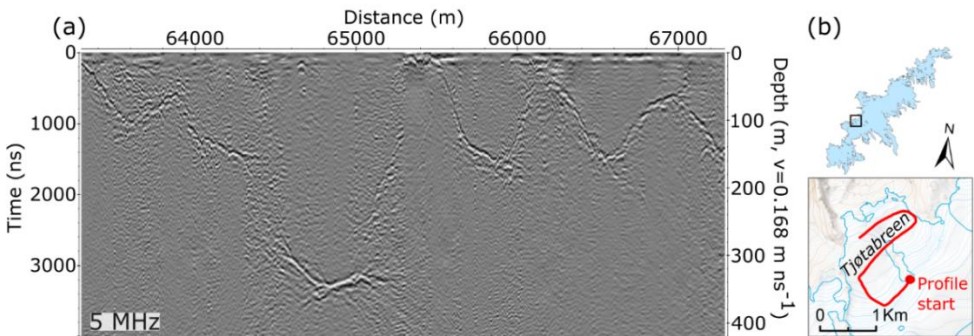

**Figure 5: (a) Example of measurements with the 5 MHz airborne radar system. (b) The profile was located along a transect at Tjøtabreen (Fig. 1). The background map in (b) is from the Norwegian Mapping Authority (WMS for**
**Topografisk Norgeskart available at https://www.geonorge.no/) and the 2019 glacier outline is from Andreassen et al. (2022).**

After the initial ice thickness calculations, all observations of ice thickness were plotted in ArcGIS Pro, where we deleted points collected with the 5 and 2.5 MHz radar systems in sharp turns, as the long antennas were not fully

extended in these locations. Profile lines collected alongside and in close proximity to valley walls were also removed to limit the influence of off-nadir reflections in the dataset. In marginal regions with both high- and ultra-high frequency observations, high-frequency measurements (2.5 and 5 MHz) were deleted due to their comparably lower accuracy. In order to produce a consistent dataset of ice thicknesses for the entire Jostedalsbreen, we double-checked interpretations at all locations where ice thickness observations from crossing profiles differed by

more than 15 m. When contrasting observations suggested that a transect was influenced by off-nadir reflectors or other uncertainties such as resolution issues, the presence of multiple reflectors or location uncertainties, these datapoints were removed from the dataset. The combination of multiple frequency measurements in many regions



of the ice cap has resulted in a dataset where both thin and very thick ice is represented in a generally satisfactory resolution (Fig. 6).


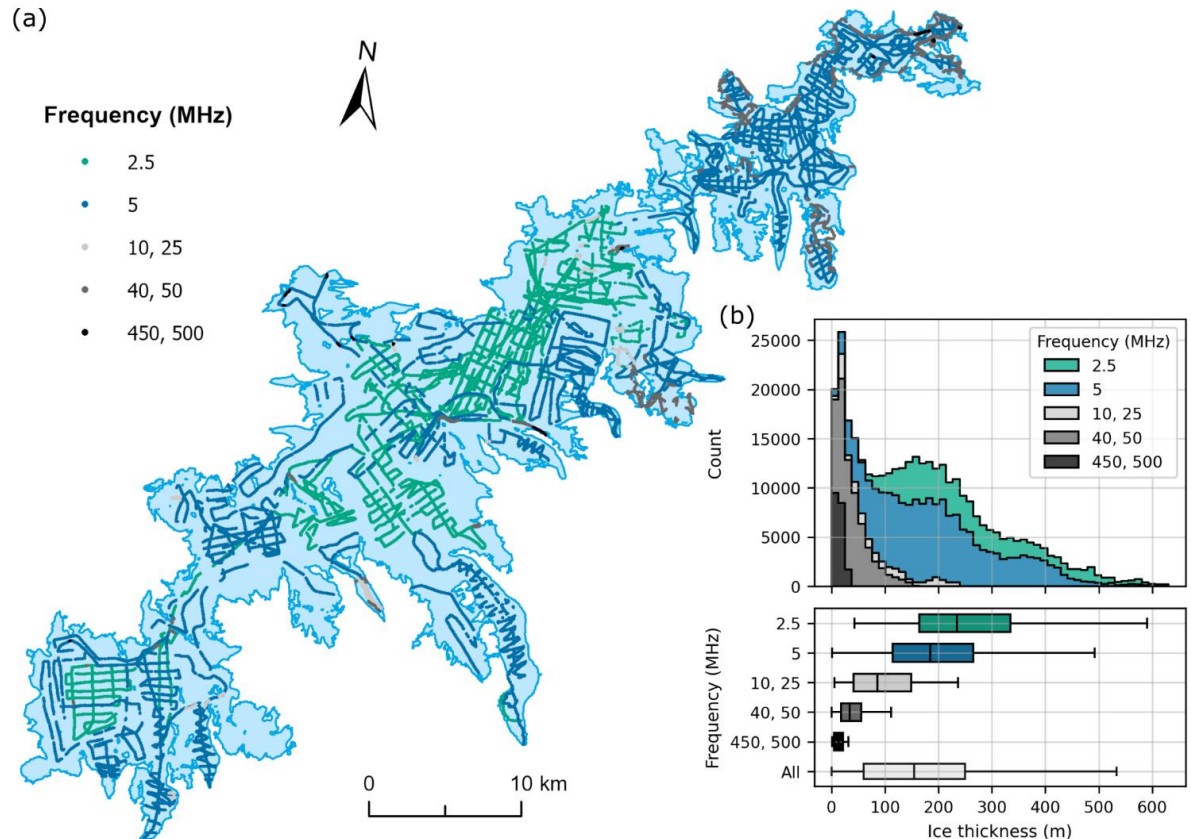

**Figure 6: (a) Ice thickness measurements across Jostedalsbreen categorized according to antenna frequency. The thickest regions of the ice cap were measured using the lowest frequency antennas, while higher frequencies were applied in the more marginal and thinner regions. (b) Histogram (top) and boxplot (bottom) of measurements of ice thickness categorised by antenna frequency. Boxes represent the interquartile range (IQR; the spread of the middle 50 % of the data), with medians indicated by vertical lines. Whiskers extend to the highest and lowest values that are within the 1.5*IQR limits. The analysis shows that measurements collected using higher frequency GPR systems dominate at low ice thickness, while 5 and 2.5 MHz GPR systems were the better choice for ice thicknesses above ~100 m.**

## 3.3 Homogenization to 2020 DTM and calculation of glacier bed topography

Following the data processing and interpretation of the GPR measurements, the bed topography elevation beneath Jostedalsbreen was calculated from the point values of ice thickness and a recent 10 m national digital terrain model (DTM10) from the Norwegian Mapping Authority. For Jostedalsbreen, the DTM10 is derived from airborne laser scanning (lidar) collected by Terratec over a seven-day period in August 2020, that covered Jostedalsbreen



and surrounding area with a point density of minimum 2 pp m$^{-2}$ (Terractec, 2020). The central part of the ice cap
was scanned on 9 August, the western part on 10 August and the eastern part on 15 August. The accuracy of the
final point cloud is assumed to be ±0.1 m (Andreassen et al., 2023). The 2020 survey (2020 DTM) covers the entire
Jostedalsbreen, except for the lower tongue of Tunsbergdalsbreen (Andreassen et al., 2023) where surface
elevation data in DTM10 is derived from stereophotogrammetry using 2017 orthophotos.


To prevent discontinuities in the elevation of bed topography, all ice thickness measurements were homogenised
to correspond to the date of the 2020 DTM. We used DGNSS observations of surface elevation to calculate an
area dependent mean surface elevation difference between the time of acquisition of GPR data and the 2020 DTM.
Calculations show that DGNSS measurements exceed the DTM by average values ranging from 0.6 m (northern
parts in spring 2022) to 3.9 m (central parts in spring 2018), reflecting surface changes such as the increased depth
of the snowpack during spring measurements compared to the end of summer lidar scan. The elevation of the bed
topography was calculated by subtracting the homogenised ice thicknesses from the 2020 DTM.

**3.4 Ice thickness measurement uncertainties**

The multifrequency dataset of crossing profiles allows for an investigation of discrepancies between measurements
with various degrees of vertical resolution as a means to evaluate ice thickness uncertainties. Here, we present the
results of a comparison of ice thicknesses at intersection points (crossover analysis), in addition to the total
calculated measurement uncertainty for each datapoint following the method described by Lapazaran et al. (2016).
In the final dataset, profiles crossed at 1207 locations (not counting profiles collected along identical tracks). Ice
thicknesses in crossing points had a mean absolute difference (MD) of 6.8 m with a standard deviation (SD) of 5.8
m, which when expressed in relation to ice thickness equals a MD of 5.0 % (7.1 % SD). Not surprisingly, the
discrepancy between values increased with decreasing frequency and hence vertical and horizontal resolution.
The largest discrepancies were observed where at least one of the crossing profiles was collected with 2.5 MHz
antennas (MD of 8.4 m and a 6.7 m SD; maximum discrepancy of 39 m; n=538), whereas profiles collected with
500 and 450 MHz antennas generally corresponded better with other observations (MD of 3.7 m and a 3.1 m SD;
maximum discrepancy of 10 m; n=23). The crossover analysis also facilitated an assessment of the performance
of the lowest frequency measurements when compared to higher resolution and more accurate ice thickness
observations collected using antenna frequencies of 25–500 MHz. The comparison show that ice thicknesses
measured with 2.5 and 5 MHz antennas were generally (but not always) somewhat larger than those measured
with higher frequency antennas. The ice thicknesses measured with 2.5 and 5 MHz antennas were on average 8.0
m (6.9 m SD; n=31) and 3.6 m (4.8 m SD; n=136) greater, respectively, than those measured with the 25–500 MHz
antennas. It is unclear exactly why these differences occur. Although a systematic bias is unfortunate, the observed



differences are well below the vertical resolution (evaluated conservatively as ½ wavelength, λ) of both the 2.5 MHz (33.6 m) and 5 MHz (16.8 m) antennas, as well as the total calculated measurement uncertainty described below.

To evaluate the performance of the new 5 MHz helicopter system, we compared discrepancies between ice thicknesses measured at intersecting airborne and ground-based profiles. We found an MD of 7.2 m (4.6 m SD; n=56) between airborne and ground-based ice thickness measurements, which is comparable to values found for all ground-based and crossing 5 MHz profiles (MD of 6.5 m and a 5.0 m SD; n=705). It is worth noting that helicopter measurements along several outlet glaciers and at steep ice falls were conducted along centreline profiles, where

off-nadir reflectors may affect the results (Fig. 1c). This could result in an underestimation of ice thickness in these regions. Where measurements along cross profiles suggested that the centreline values were unreliable, the latter were removed from the dataset. However, in most cases centreline values compared well with measurements along cross profiles and were largely included in the dataset.

As a crossover analysis does not encompass all potential uncertainties associated with ice thickness measurements, it is generally considered to only provide a rough approximation of uncertainty (Lapazaran et al., 2016). Consequently, we calculated the total measurement uncertainty for each ice thickness observation using the method described by Lapazaran et al. (2016), which is based on the root-sum-of-squares of both uncertainties in the ice thickness measurements and the measurement position. Using this approach, we included uncertainties

related to the radio-wave velocity, which we assumed to be 5 %, as recommended by Lapazaran et al. (2016) when the same velocity is applied in both accumulation and ablation areas. In addition, our uncertainty calculations considered the signal resolution (λ/2) and positioning uncertainty. The latter was accounted for by calculating the largest measured ice thickness difference within a circle, with the radius determined by the respective GNSS uncertainty. Using this approach, total ice thickness uncertainties were primarily controlled by antenna frequency

and ice thickness because of their influences on vertical resolution and the uncertainty caused by the constant radio-wave velocity, respectively (Fig. 7 and Fig. B1).

The calculated combined uncertainties of the ice thickness measurements amounted to an average of 19.6 m for the entire dataset (SD of 12.1 m; n = 351 559), while mean ice thickness uncertainties ranged between 36.5 m (SD

of 2.5 m) and 20.2 m (SD of 3.1 m) for 2.5 and 5 MHz measurements, respectively, and 1 m (SD of 0.5 m) for 450 and 500 MHz measurements. The large mean uncertainty estimate calculated for most ice thickness observations was primarily a result of the conservative treatment of signal resolution and the assumed 5 % uncertainty from applying a single radio-wave velocity value to the entire ice cap despite ice cap-wide variations in snow, firn, and thermal ice conditions. The significantly larger measurement uncertainty found using the method of Lapazaran et

al. (2016) compared to the crossover analysis (Fig. 7b), implies that the former approach leads to an overestimation



of uncertainties associated with relatively low frequency (below ~10 MHz) ice thickness measurements, particularly in regions with thick ice. We therefore suggest that the crossover analysis and the calculated measurement uncertainty represent a lower and upper estimate, respectively, of the uncertainties associated with each ice thickness observation. In the datafile compilation presented here, we include only the upper estimate of total

measurement uncertainty.

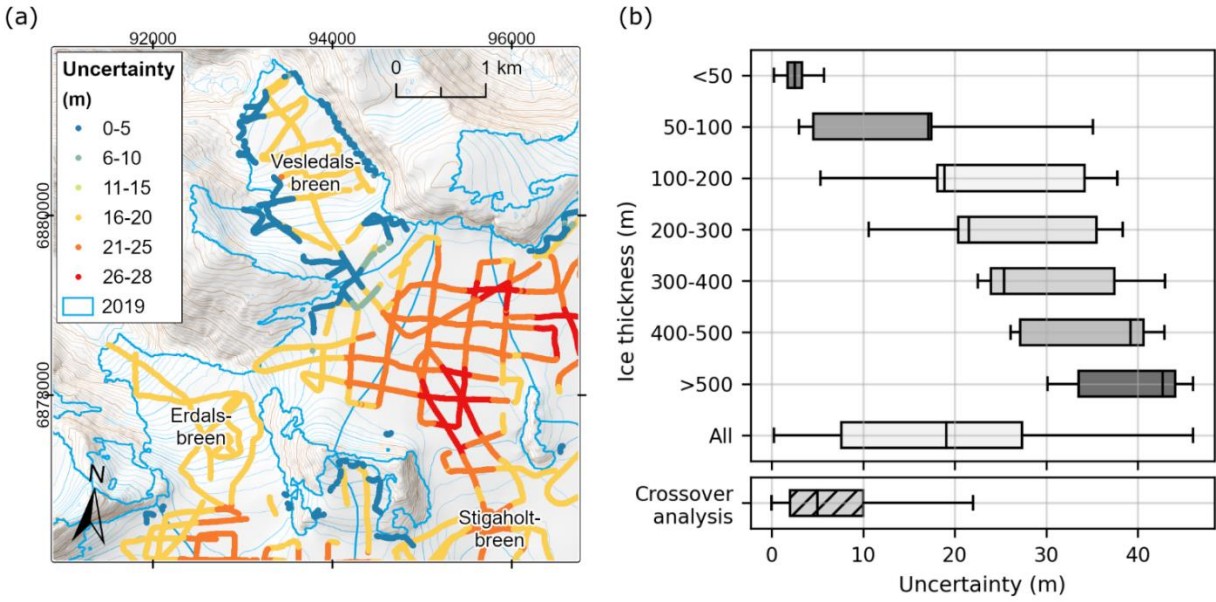

**Figure 7: (a) Calculated ice thickness measurement uncertainties at Vesledalsbreen (Fig. 1). Variations in measurement uncertainties are primarily controlled by antenna frequency, with <5 m uncertainty for 500 MHz measurements,**
**between 6 and 13 m uncertainty for 50 MHz measurements and ≥14 m for 5 MHz measurements. The largest measurement uncertainties are found in regions with thick ice, illustrating the influence of ice thickness on the uncertainty calculations. (b) Distribution of calculated absolute uncertainty in ice thickness by thickness class and for all measurements following the method described by Lapazaran et al. (2016), as well as that observed in the crossover analysis. Boxes represent the interquartile range (IQR; the spread of the middle 50 % of the data), with medians**
**indicated by vertical lines. Whiskers extend to the highest and lowest values that are within the 1.5*IQR limits. The background map in (a) is from the Norwegian Mapping Authority (WMS for Topografisk Norgeskart available at https://www.geonorge.no/) and the 2019 outline of Jostedalsbreen glacier units is from Andreassen et al. (2022). The coordinate system is UTM 33N, datum ETRS_1989.**

### 3.5 Description of datafile compilation

The ice thickness point values from Jostedalsbreen were compiled in a format similar to that of the Glacier Thickness Database (GlaThiDa Consortium, 2020; Welty et al., 2020) for straight-forward application in future studies. Data were stored in a CSV (comma-separated values) file with attributes describing the data (Table 2), and a DOI is provided for the ice thickness dataset. Consequently, the dataset follows the FAIR principles of optimised findability, accessibility, interoperability, and reusability.




**Table 2: Attributes used in the point dataset of ice thickness values on Jostedalsbreen.**

| Attributed field | Unit | Description |
|---|---|---|
| SURVEY_DATE | YYYYMMDD | Survey date |
| PROFILE_ID | Text | Identifier of processed radar profile |
| POINT_ID | Number: 1-n | Point identifier |
| ANTENNA_FREQUENCY | MHz | Antenna frequency of measurement |
| SURVEY_METHOD | Text: H, S or F | Means of transport during survey (H: Helicopter, S: Scooter, F: Foot) |
| GNSS_SOURCE | Number: 0 or 1 | Position information (0: Radar GNSS (lowest uncertainty) and 1: External GNSS source or some degree of interpolation across minor data gaps) |
| POINT_LAT | DDD.DDDDDD° | Latitude of point value |
| POINT_LON | DDD.DDDDDD° | Longitude of point values |
| GNSS_ELEVATION | m a.s.l. | Surface elevation from GPR GNSS |
| THICKNESS | Meter | Ice thickness value |
| THICKNESS_UNCERTAINTY | Meter | Uncertainty in ice thickness based on Lapazaran et al. (2016) |
| THICKNESS_2020DTM | Meter | Ice thickness value homogenised to the 2020 DTM surface. Corrected for differences in surface elevation during survey years relative to the 2020 DTM. |

*Survey date August 2020 except for the lower part of Tunsbergdalsbreen.

Most of the attributes in the table containing ice thickness point values are self-explanatory and identical to those
in GlaThiDa. However, data entries such as SURVEY_METHOD, GNSS_SOURCE and THICKNESS_2020DTM
are additional attributes to describe the Jostedalsbreen data collection. In addition to the datafile containing the
complete ice thickness dataset (n = 351 559 entries), we provide a thinned-out version of this dataset (n = 35 100
entries) consisting of point values extracted randomly from the full dataset but with a minimum distance of 20 m.
The smaller dataset allows for easier plotting and analysis.

**3.6 Model-based ice thickness extrapolation**

While the dense network of GPR profiles across large parts of the ice cap provides direct local information on ice
thickness on 59 out of the 81 glacier units that make up Jostedalsbreen ice cap (Fig. 1), an extrapolation to
unmeasured regions was necessary to produce grids of ice thickness and bed topography which cover the entire
Jostedalsbreen. Here, we apply an approach that combines the advantages of inter- and extrapolation of ice
thickness observations with those of ice thickness modelling from an inversion of surface topography (Huss and



Farinotti, 2014; Grab et al., 2021). The basis of this approach is an ice thickness model originally developed for global-scale applications (Huss and Farinotti, 2012). The model was used in the Ice Thickness Model Intercomparison eXperiment (ITMIX and ITMIX2, Farinotti et al., 2017, 2021) and performed well in estimations of ice thickness distribution and bed topography. The general concept of the model is to derive local ice thickness
from surface characteristics. It relies on glacier surface hypsometry of all individual glacier units of Jostedalsbreen, discretised into 10 m elevation bands. Variations in the valley shape and the basal shear stress along each outlet glacier's longitudinal profile, as well as an estimated longitudinal trend in basal sliding (e.g., Huss and Farinotti, 2012), are taken into account. Ice volume fluxes are computed along a longitudinal profile based on calibrated mass balance gradients. Subsequently, ice thickness is calculated by inverting the flow law for ice (Glen, 1955).
Resulting averages of elevation-band ice thickness are then extrapolated to a regular grid by considering both local surface slope and distance from the glacier margin, excluding ice divides (for details see Huss and Farinotti, 2012).

Before initialising the model-based ice thickness extrapolation, we harmonised the spacing of the acquired profiles by taking the average of all homogenised ice thickness point data contained within the same 10 x 10 m cell of the
DTM10. The ice thickness point dataset and the outline of Jostedalsbreen both serve as important input when computing spatially distributed ice thickness. As glacier outline, we used the national glacier inventory which relies on Sentinel-2 images taken on 27 August 2019 (Andreassen et al., 2022). In this dataset, Jostedalsbreen is divided into glacier units from topographic observations on ice divides. The inventory was derived using a standard semi-automatic method and checked against orthophotos and Sentinel composites from 2017 and 2019, respectively,
with manual edits to correct for areas in shadow, with debris-cover, and lake outlines. The uncertainty in the outlines of the final product was estimated to be within half a pixel (±5 m).

Our dataset of distributed ice thickness for all Jostedalsbreen was produced by optimising modelled ice thickness to local ice thickness observations for each individual glacier unit, following a three-step procedure that consisted
of (i) model optimisation, (ii) spatial bias-correction of modelled thicknesses, and (iii) spatial interpolation relying on point values of thickness and bias-corrected model results for regions that are not covered by GPR surveys.

In step (i), we optimised the apparent mass-balance gradient (Farinotti et al., 2009) in an automatic procedure to minimise the average misfit between modelled ice thickness and the available observations for each of the 59 outlet
glaciers with ice thickness measurements. The apparent mass balance was then computed based on two linear elevation gradients, one for the ablation area and one for the accumulation area, assuming a balanced mass budget for the entire glacier unit. The resulting apparent mass balance distribution was then used to compute ice volume fluxes from the top to the bottom of each glacier unit, and to infer modelled ice thickness distribution as in Andreassen et al. (2015).






In step (ii), the modelled ice thickness distribution from step (i) was bias-corrected using ice thickness point values. First, relative differences between modelled and measured point ice thickness distributions were evaluated. These differences were then spatially inter- and extrapolated based on an inverse-distance weighting scheme. This relative spatial ice thickness correction field was then superimposed on the modelled ice thickness distribution,

resulting in a bias-corrected model-based ice thickness distribution that accounts for the differences between observed and modelled ice thickness at a spatially distributed scale. Nevertheless, this ice thickness distribution will not exactly match all GPR-derived point values of thickness.

In the final step (iii), we spatially interpolated the ice thickness distribution based on (1) all available ice thickness

observations, (2) the model results adjusted in steps (i) and (ii) in regions that were not covered by direct measurements (buffered in a distance of 100–200 m around available observations depending on outlet glacier size), and (3) the condition of zero ice thickness on the glacier margin, except for ice divides. The ice thickness at ice divides was obtained from model results of neighbouring outlet glaciers, and also entered the interpolation. Furthermore, a set of individually estimated thicknesses on ice divides based on local knowledge and direct

interpolation of nearby GPR profiles was included to increase the robustness of spatially complete ice thickness estimates at ice divides. Repeating the complete procedure several times ensured convergence and thus consistency of thicknesses on both sides of the ice divides. For glacier units without GPR measurements, the ice thickness model was run using average calibrated parameters of the apparent mass-balance gradient from all outlet glaciers with direct observations. This direct modelling of ice thickness, however, was only relevant for small

and mostly thin glacier units within Jostedalsbreen, and account for just 1.9 % of the total inferred volume of the ice cap. We finally combined all results of extrapolated ice thickness from the 81 glacier units contained in Jostedalsbreen into a complete coverage with a spatial resolution of 10 x 10 m.

### 3.7 Bed topography and potential future lakes

Bed topography was obtained by subtracting distributed ice thickness from the DTM10 ice surface elevation. The

resulting grid of bed topography was then smoothed with a spatial filter of 50–100 m (depending on glacier basin area) to remove remaining discontinuities at ice divides, as well as unrealistic small-scale variability in calculated bed topography that cannot be inferred with the applied methodology and will originate from surface features. Depressions in the bed topography might act as potential future lakes after complete disappearance of the ice cover. Even though the uncertainty in detecting the extent and volume of such depressions is large, we derived a

map of potential lake area and depth from the map of subglacial bed topography. This was achieved by using a sink fill algorithm that detected depressions, after which the depth and volume of each depression was determined





by artificially filling the depression until they overflow. This resulted in an inventory of individual potential glacier lakes, including the relevant attributes, such as their elevation, area, volume, or maximum depth.

### 3.8 Uncertainties in extrapolated ice thickness

The uncertainty in extrapolated ice thickness is composed of two elements: (1) the uncertainty in measured ice thickness, and (2) the uncertainty induced when extrapolating point ice thickness across the entire ice cap supported by the model-based approach. These two elements of uncertainty are estimated separately, and then propagated through the methodology described above to derive a spatially distributed uncertainty map for the entire ice cap. As described in section 3.4, the uncertainty associated with each point value of ice thickness was calculated

following Laparazan et al. (2016). We conservatively assume all uncertainties across the entire ice cap to be correlated and generate a dataset with maximum observed thickness and minimum ice thickness according to the above uncertainties. Based on these two datasets, we repeated the approach described in section 3.6 using each of these datasets. Taking the mean local deviation of the results from the ice thickness distribution inferred with the reference approach, we computed a spatially distributed uncertainty estimate due to measurement uncertainty.


To assess the uncertainty caused by extrapolating observations to unmeasured regions, we performed a suite of sensitivity experiments by varying different parameters of the model-based approach within conservatively set, but physically meaningful, ranges. This was performed for the viscosity of ice, the assumed fraction of basal sliding, and the apparent mass balance gradients. In each experiment, the reference dataset of point ice thickness values

was used for calibration, such that the resulting ice thickness grids differ mostly in regions where ice thickness is solely inferred by the model.

Finally, we combined the local offset from the reference ice thickness distribution for all experiments based on the root-sum-of-squares resulting in an absolute and a relative uncertainty grid (Fig. 8). Local uncertainties were

bounded to not exceed the grid cell's reference ice thickness which occurred in a few instances close to glacier margins. Typically, this grid indicates small uncertainties close to the GPR profiles and larger uncertainties in regions where the result is based on ice thickness modelling. Overall, we find a mean uncertainty in local ice thickness of 36 m (30 %), where regions with thick ice are characterised by high absolute but low relative thickness uncertainties, and vice versa for regions with thin ice (Fig. 8).








Figure 8: (a) Absolute and (b) relative uncertainty for distributed ice thickness on Jostedalsbreen. The two figures illustrate that the largest absolute uncertainties appear in regions with thick ice and away from GPR profiles, while the largest relative uncertainties are found in the thin ice marginal regions. The 2019 outline of Jostedalsbreen glacier units is from Andreassen et al. (2022).




# 4 Results

## 4.1 Measurements of ice thickness

The dataset presented here provides ice thickness point values for 59 of the 81 glacier units that constitute the Jostedalsbreen 2019 inventory. These 59 glaciers cover 437 km², or 95 % of the total area of the ice cap (458 km²
in 2019). All parts of Jostedalsbreen are now less than 900 m from a point of known ice thickness (measurement or glacier outline), while distances to a known point are less than 300 m for 90 % of the ice cap and less than 100 m for 49 % of the ice cap. A maximum ice thickness of 631 m (or 628 m when referring to 2020 DTM) was measured in the upper accumulation area of Tunsbergdalsbreen, which is the largest outlet glacier of Jostedalsbreen and located in the central part of the ice cap (Fig. 4 and 9). In Jostedalsbreen South and North, ice thickness reaches
maximum values of ~520 and ~430 m, respectively. In general, the thickest ice at Jostedalsbreen is found in the flattest areas of the ice cap, while thinner ice of less than 100 m thickness covers protruding hills. In the northern parts, the highest mountains in the landscape surrounding Stigaholtbreen (Fig. 7 and 9) are already partially ice-free, giving the ice cap a more disjointed appearance in this region.

Along the south-eastern margin of Jostedalsbreen, large outlet glaciers flow far into the valleys below. Particularly thick ice is found along the three glacier tongues of Tunsbergdalsbreen (up to ~615 m), Flatbreen (up to ~435 m) and Stigaholtbreen (up to ~320 m) (Fig. 9). These outlet glaciers are characterised by large accumulation areas from which ice flows relatively unrestricted from the innermost parts of the ice cap plateau and along deep glacier-carved valleys. In comparison, thinner ice is observed along outlet glaciers where ice flows from the ice cap plateau
through steep ice falls. Austerdalsbreen with its two steep ice falls and low-sloping glacier tongue, represents one such example. Here, helicopter measurements along the centre flowline of the largest of the two narrow ice falls suggest that the ice is only 40–50 m thick in the steepest parts. Below the ice falls, ice thickness reaches a maximum of ~235 m. At Nigardsbreen, ice also thins to 40–50 m as it flows through the two smallest western ice falls. Here, the main flow of ice from the ice cap plateau appears to occur through the much larger northern tributary,
where centre-line ice thicknesses of more than 100 m were measured in the thinnest regions. Below the three ice falls, ice thickness reaches a maximum of ~265 m before thinning towards the famous glacier front of Nigardsbreen.



**Figure 9: (a) Combined ice thickness observations from the field campaigns in 2018, 2021, 2022 and 2023. The point of maximum thickness is marked with a red triangle. (b) Section of Lodalsbreen with 100 m surface contours. Note that the helicopter measurements along Lodalsbreen were collected during the first test flight of the airborne radar system, where profile locations were positioned less than ideal in relation to the valley orientation. The background mountain shadow and 100 m contour lines in (b) are from the Norwegian Mapping Authority (WMS for Topografisk Norgeskart available at https://www.geonorge.no/). The 2019 outline of Jostedalsbreen glacier units is from Andreassen et al. (2022) and the coordinate system is UTM 33N, datum ETRS_1989.**

From the extensive measurements of ice thickness, we have identified two regions that may be particularly vulnerable to future climate-forced changes and that have the potential to separate Jostedalsbreen into three unconnected ice caps, North, Central, and South (Fig. 1). In the north, Lodalsbreen currently connects the northernmost part of Jostedalsbreen with its more southern regions through three steep tributaries (Fig. 9b). Helicopter measurements along the centre flowlines reveal that the ice thins to 50 m or less as it flows southwards and into the incised valley below. Ice flowing from the western tributary is thicker, with ice thicknesses ranging between 50 and 70 m along its thinnest sections. Further south on Jostedalsbreen, thin ice of less than 25 m covers



the narrow stretch at Grensevarden that joins the southern part of the ice cap with its central regions (Figs. 3 and 9). Bedrock has already started protruding through the thinning ice, and the emerging rocks are likely to further accelerate the changes occurring in this part of Jostedalsbreen due to positive feedback on melting from a decreasing albedo of the surroundings. However, it is important to note that while thin ice may indicate increased

vulnerability to future warming, other factors such as ice velocity and surface mass balance are important influences when considering future changes in areas with thin ice. Such considerations require ice cap-wide modelling of glacier evolution and are beyond the scope of this paper.

**4.2 Comparison to previous ice thickness measurements at Jostedalsbreen**

The new comprehensive dataset of Jostedalsbreen ice thicknesses represents a significant improvement to

previous measurements, both in relation to data quality and spatial coverage across the ice cap. We now have a much better understanding of ice thickness variations in the region and have also extended the maximum measured ice thickness from 600 m measured during the 1980s field campaigns (Sætrang and Wold, 1986) to the 631 m measured in 2021. Many of the previous ice thickness measurements conducted on Jostedalsbreen have considerable uncertainties in measurement positioning and surface topography. Therefore, we limit the comparison

of our measurements to ice thickness observations on Austdalsbreen in the late 1980s, which we consider to be afflicted with the lowest uncertainties. This older dataset was collected to evaluate future changes to Austdalsbreen due to enhanced calving after the regulation of the proglacial lakes Austdalsvatnet and Styggevatnet for hydropower production (Hooke et al., 1989; Laumann and Wold, 1992). Ice thickness was measured in nine hot water drilled boreholes and by GPR within an area of 600 by 1000 m, where the ice thicknesses ranged between

100 and 230 m (Sætrang and Holmqvist, 1987; Sætrang, 1988). The boreholes were drilled in September 1986 and October 1987, while the GPR measurements used here for the assessment of uncertainties were collected in April–May 1988 using an 8 MHz radar system. Comparisons between radar measurements and boreholes at the time showed borehole bedrock elevations between 14 m below and 1 m above radar bed elevations. The overall uncertainty of the radar bed elevations was estimated to be within 7 m based on results from a radar crossover

analysis and observed uncertainties in positioning and surface elevation (Sætrang, 1988).

Two radar profiles from 2022 intersected the area also mapped by GPR in 1988. To allow for a comparison with the new ice thickness measurements, we interpolated a 5 x 5 m bed elevation grid from the 1988 GPR measurements and extracted the bed elevations at the nine boreholes and 454 locations covered by the GPR

survey in 2022. On average, bed elevations measured in boreholes were 4 m lower than the interpolated grid, and the grid consequently shows a good replication of variations observed in both of the two older datasets. When comparing values from the interpolated grid and those obtained in 2022, we find that bed elevations calculated





from measurements in 2022 were on average 14 m lower than those found with GPR in 1988 (i.e., 2022 ice was thicker than expected from the 1988 dataset). However, it is unclear whether this discrepancy relates to uncertainties concerning the earlier or the new measurements. In this region the 2022 measurements have a measurement uncertainty of 17–20 m (Fig. B1), and the observed discrepancies are consequently within the range of combined uncertainties.

### 4.3 Distributed ice thickness, bed topography and potential future lakes

The maps of ice thickness and bed topography (Fig. 10) allow for a coherent description of the variations in the morphology of Jostedalsbreen, also in regions that are not covered by GPR measurements. The two grids illustrate that thickest ice is found predominantly away from ice divides and in the prominent subglacial valleys of the largest outlet glaciers. By contrast, thinner ice and elevated subglacial bed topography are often associated with regions of the ice cap with high surface elevations. From the modelled ice thickness grid, we calculate an ice cap-wide mean ice thickness of 154 m ±22 m and a present (~2020) ice volume of 70.6 ±10.2 km$^3$ (Table 3). Overall, the presented results are consistent with previous estimates for Jostedalsbreen, and any smaller discrepancies are well within the uncertainty of the applied methodologies. The calculated mean ice thickness is slightly smaller than the earlier estimate of 158 m which was calculated for an interpolated region covering 65 % (310 km$^2$) of the 2006 area (474 km$^2$) of Jostedalsbreen (Andreassen et al., 2015). Our calculated ice volume also compares well with the estimate of 72.6 km$^3$ provided by Frank and van Pelt (2024).





Figure 10: (a) Modelled distributed 10 m ice thickness of Jostedalsbreen and (b) distributed 10 m bed calculated from DTM10 and the modelled ice thickness distribution (Fig. 10a). The 2019 outline of Jostedalsbreen glacier units is from Andreassen et al. (2022).






Calculations of key numbers for selected elements of the ice cap (Table 3) show that Jostedalsbreen Central is by far the largest of the three regions when comparing area, mean ice thickness and volume. The two surrounding regions have much smaller areas and ice is generally thinner, in particularly in the smallest northernmost region.

The ice thickness measurements presented in section 4.1 illustrate the vulnerability of Jostedalsbreen to future separation into three minor ice caps. Following a future breakup, Jostedalsbreen Central would remain the largest glacier in Norway and mainland Europe, surpassing the second largest glacier, Vestre Svartisen, which had an area of 192.2 km² in 2019 (Andreassen et al., 2022).

**Table 3: Key numbers for the three regions and prominent outlet glaciers based on calculations from the model-based grid of ice thickness for Jostedalsbreen. The bracketed values after each glacier name refer to glacier IDs from Andreassen and Winsvold (2012b). Data coverage is defined as all regions which are less than 300 m from a point of known ice thickness (measurements or glacier outline), with bracketed values specifying the percentage of the area which are less than 100 m from a known point.**

| Glacier | Area (km²) | Maximum (m) | Mean (m) | Volume (km³) | Data coverage (%) |
|---|---|---|---|---|---|
| Jostedalsbreen | 458.1 | 626 | 154 | 70.6 | 90 (49) |
| North | 69.3 | 432 | 123 | 8.5 | 99 (69) |
| Central | 309.6 | 626 | 161 | 49.9 | 88 (45) |
| South | 79.3 | 518 | 155 | 12.3 | 91 (47) |
| Lodalsbreen (2266) | 8.8 | 329 | 93 | 0.88 | 98 (57) |
| Kjenndalsbreen (2296) | 19.1 | 419 | 186 | 3.6 | 92 (50) |
| Nigardsbreen (2297) | 41.7 | 572 | 178 | 7.4 | 98 (62) |
| Nigardsbreen MB* (2311, 2299 and 2297) | 45.4 | 572 | 169 | 7.6 | 98 (62) |
| Tunsbergdalsbreen (2320) | 46.2 | 626 | 233 | 10.8 | 89 (45) |
| Austerdalsbreen (2327) | 19.4 | 510 | 191 | 3.7 | 85 (44) |
| Bøyabreen (2349) | 13.8 | 501 | 201 | 2.8 | 99 (53) |
| Flatbreen/Supphellebreen (2352) | 12.7 | 452 | 205 | 2.68 | 97 (58) |
| Austdalsbreen (2478) | 10.3 | 402 | 188 | 1.98 | 100 (70) |
| Stigaholtbreen (2480) | 12.5 | 432 | 188 | 2.38 | 99 (65) |

*Nigardsbreen MB refers to the mass balance glacier basin used by Andreassen et al. (2023).

Beneath Jostedalsbreen we observe a versatile landscape of deep glacially incised valleys that extend to the centre of the ice cap in some regions, and are surrounded by steep valley walls, hanging valleys and glacial over-





deepenings (Fig. 10b). The map of bed topography provides a glimpse of how the landscape would look like if

Jostedalsbreen was to completely disappear and from it we can infer possible future changes in the regional hydrological systems. While a detailed analysis of hydrological changes in the region is outside the scope of this study, it is worth noting that several glaciers have discrepancies between the ice divides defined by the current surface topography of the ice cap and the hydrological catchment boundaries determined by the bed topography in an ice-free landscape. Examples of such are Flatbreen (Supphellebreen), Tunsbergsdalsbreen and

Nigardsbreen, where the subglacial valleys appear to extend significantly beyond the current ice divides (Fig. 10b). Other glaciers, such as at Austerdalsbreen and Lodalsbreen, have similar surface and subglacial topographical divides. Overall, it appears likely that in an ice-free landscape, upper catchment boundaries in the central and southern Jostedalsbreen regions will, in many places, be located further north and northwest than the currently more central longitudinal ice divide. In the northern parts of Jostedalsbreen, the potential extent of ice-free

catchment areas appears more uncertain due to several smaller thresholds in the bed topography and limitations in data coverage across these. Consequently, we tentatively suggest that in an ice-free landscape, the topographic bed catchment at Austdalsbreen may increase substantially in size at the expense of the surrounding regions, although further analysis is required to substantiate this claim.

The distributed bed topography furthermore reveals subglacial bed depressions as likely locations for future lakes in a warming climate (Fig. 11). Our results show a multitude of potential lakes, the largest of which is 3.5 km long and has an area of 2.4 km$^2$ and is located in the inner regions of Tunsbergdalsbreen, just south of where the thickest ice was measured. Other large topographic depressions are found north of Bøyabreen and Flatbreen glacier fronts, underneath the glacier tongue of Tunsbergdalsbreen, and north-west of the calving front of

Austdalsbreen. According to our estimates, a total of 14 % (65.3 km$^2$) of the present-day glacier area of 458 km$^2$ (2019) can be covered by lakes if the entire Jostedalsbreen melts away.





**Figure 11: Location of current and potential future lakes calculated from the grid of subglacial bed topography at**
**Jostedalsbreen (Fig. 10b). The largest potential future lake is marked by a red triangle. The 2019 outline of Jostedalsbreen glacier is from Andreassen et al. (2022) and the background mountain shadow and outline is from the Norwegian Mapping Authority. Outline of present-day lakes is from the Norwegian Mapping Authority (WMS for Topografisk Norgeskart available at https://www.geonorge.no/) and the Norwegian Water Resources and Energy Directorate (https://doi.org/10.1017/jog.2022.20). The coordinate system is UTM 33N, datum ETRS_1989.**






## 5 Data availability

All ice thickness observations (complete and thinned-out compilations) and maps of ice cap-wide ice thickness, combined uncertainty in ice thickness, bed topography and outlines of potential future lakes are available for download at https://doi.org/10.58059/yhwr-rx55 which is hosted by the Norwegian Nasjonalt Vitenarkiv (Gillespie
et al., 2024).

## 6 Conclusions

In this paper we present a rich point dataset of high-quality ice thickness observations on Jostedalsbreen ice cap collected during GPR surveys in 2018–2023. Measurements were collected from 59 of the 81 glacier units that constitute Jostedalsbreen and 90 % of the total ice cap area is now less than 300 m from a point of known ice
thickness. A maximum ice thickness of ~630 m was measured on Tunsbergdalsbreen outlet glacier in the central part of the ice cap. This measurement exceeds the 600 m maximum thickness previously measured on Jostedalsbreen (Sætrang and Wold, 1986; Andreassen et al., 2015). Smaller maximum ice thicknesses of ~520 m and ~430 m were measured in the southern and northern parts of the ice cap, respectively. Using this new dataset of ice thickness values, we produce model-based grids of distributed ice thickness and bed topography that allow
for a coherent description of ice thickness variations and subglacial morphology over the entire Jostedalsbreen, as well as calculations of key figures for the ice cap. We find that Jostedalsbreen has a mean thickness of 154 m ±22 m and a present (~2020) ice volume of 70.6 ±10.2 km$^3$. Together, the ice thickness measurements and distributed datasets provide exceptional new details about the geometry and bed topography of Jostedalsbreen, revealing vulnerabilities to future ice cap fragmentation and possible changes in the hydrological systems with climate
warming. These datasets will form the basis of future studies of climate-induced changes in the Jostedalsbreen region, which are of high importance to local stakeholders such as farmers, tourist operators and hydropower companies.

## Author contributions

MKG, JCY, and LMA designed the study. MKG led the data collection of ice thickness measurements and MKG,
SDV, KHS, JA, JB, JMC, HE, BK, EL, MM, KM, SDN, TOR, EWNS and KØ carried out the fieldwork. MKG subsequently processed and interpreted the ice thickness data. MH ran the model-based extrapolation of ice thickness measurements and prepared all distributed datasets while MKG, LMA and KHS produced the figures. MKG, LMA and MH prepared the manuscript with contributions from all co-authors. JCY was the principal investigator of the JOSTICE project.




## Acknowledgements

We would like to express our sincere gratitude to all who have contributed to the planning and implementation of the comprehensive and challenging fieldwork that was required to adequately map ice thickness across Jostedalsbreen. Especially, we would like to thank Ingebjørg Haugland for assisting during the 2021 field campaign
and Jostedalsbreen National Park, Nigardsbreen Nature Reserve, Breheimen National Park and the municipalities of Luster, Stryn, Sogndal, Sunnfjord and Skjåk who all granted permissions for the fieldwork. We would also like to thank Airlift AS who provided logistical support during both ground-based and airborne radar surveys. Steinmannen and Statkraft-hytta mountain huts, both owned by Statkraft, generously accommodated us during the fieldwork and snowmobiles were provided by Vang Auto-Service AS, Luster Red Cross mountain rescue group and Statkraft.
Lastly, we thank Statkraft for advising on weather conditions and NVE for their local avalanche forecasting.

## Competing interests

All co-authors other than EL declare that they have no conflict of interest. EL works for the hydropower company Statkraft, and Statkraft has an interest in the hydropower production at Austdalsbreen. Statkraft did not in any way influence the research objectives, data collection, analysis or interpretations of data presented in this paper.

## 730 Financial support

This study is a contribution to the JOSTICE project funded by the Norwegian Research Council (RCN grant #302458). In addition, the 2023 airborne survey was supported by funding from UH-nett Vest.

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




## Appendices

## Appendix A



**Figure A1: Locations if ice thickness measurements divided into survey year. The coordinate system is UTM 33N,**
**datum ETRS89.**





## Appendix B



**Figure B1:** **Total measurement uncertainty associated with each ice thickness observation calculated using the method described by Lapazaran et al. (2016). The coordinate system is UTM 33N, datum ETRS89.**