# Peer review of "Ice thickness and bed topography of Jostedalsbreen ice cap, Norway"

_Earth System Science Data, 2024_

## Referee Comment (RC2)

Summary and comments on the manuscript
essd-2024-167 entitled

**Ice thickness and bed topography of Jostedalsbreen ice cap, Norway**

presented on 06.05.2024
by

Mette Gillespie et al.

**SUMMARY**

The authors present an extensive record of ground penetrating radar
(GPR) measurements to map the glacier ice thickness of Jostedalsbreen
collected during multiple ground and airborne field campaigns between
2018 and 2023.  Data was collected with various radar systems involving
operation frequencies between 2.5 and 500 MHz.  The acquired record
is impressive with more than 350'000 point measurements and more
than 1'000 line kilometers surveyed.  These measurements are ingested
into an inverse procedure to infer a high-resolution map of glacier
ice thickness for the entire glacierised area.  This map is truly
valuable, as previous reconstructions of glacier ice thickness had
no measurements at their disposal.  Moreover, the authors provide
an uncertainty map of their glacier ice-thickness field.  A brief
analysis of the basal topography for sub-glacial over-deepenings
- i.e., possible sites for future lake formation - completes the
manuscript.

I was very excited about this article and I want to admit that I
was at no point disappointed.  I want to congratulate the authors
to this piece of work.  The manuscript is very well written and
strikes with clearness and high-quality illustrations.  Below you
find some major comments on several aspects of the methods and analysis.
None of them are fundamental but will help to improve or to better
assess your results.  Overall, I am very positive about this manuscript
because it combines 'impressively extensive' field observations
with modelling techniques.  I therefore recommend that the editor
should continue to considered this manuscript for publication in
Earth System Science Data after minor revisions.

**MAJOR COMMENTS**

**OPTIMAL RECONSTRUCTION**

You use an approach by Huss and Farinotti (2012) (HF apporach) to infer the glacier ice thickness of Jostedahlsbreen. Several parameters in this approach are not well constrained. How did you select the optimal parameters with regard to ice-dynamics as well as mass overturning/surface mass balance. Couldn't you use your abundant measurement record for a dedicated calibration.

**THICKNESS HOMOGENISATION**

As you have a second DEM from 2017 (aside the 2020 DEM) you could infer an elevation change map (possible a co-registration is necessary). So you could refrain from using individual elevation differences from DGNSS measurements to homogenise your thickness data. In this way, you account for spatial difference in elevation change. An elevation change map would further be useful for my comment on the 'future assessment' (below).

**FUTURE ASSESSMENT**

I think for the potential disconnection of Jostedalsbreen (L574-587), you have to combine your thickness observations with actual elevation changes observed by satellite remote sensing. I say that because even thin ice can prevail for a long time at high elevation. Moreover, I would rather use the thickness map to analyse future disconnection possibilities - instead of the point measurements.

**UNCERTAINTY MAP** I remain confused about how you built up this final map of thickness uncertainty (Sect. 3.8). You first produce additional thickness fields by feeding the minimum and maximum thickness estimates from your observations (relying on the measurements error) into the HF approach. For the extrapolation uncertainty, you vary certain model parameters. Yet it is unclear how many parameter combinations you tried and how you sample. You stay rather vague here. Moreover, I did not find which measure you used to quantify the variability (min/max, sigma, ...). Lastly, it is not clear to me how you combined the measurement error maps with these extrapolation uncertainty maps to produce a final uncertainty map. Please be more specific.

**DISCUSSION**

You state that the volume is very similar to previous estimates. I strongly doubt that these previous estimates did rely on as many thickness measurements as you had. So why are these estimates so similar. Do we no longer need to conduct measurements? I strongly

doubt that. I think there must be quite some differences in the
thickness distribution - worth to discuss. Did these approaches
use thickness observations in this region? GlaThiDa 3.1.0 holds
no data on Jostedalsbreen. What about a comparison with the global
products from Farinott et al. (2019) and Millan et al. (2022),
that many people do use. I think that a map comparison of ice thickness
is a worthwhile effort here.

**STRUCTURE**
In the uncertainty subsection of the extrapolated map product (3.8),
you present already quite some results. Please transfer these to
Section 4.

**MINOR COMMENTS**

**L757 :** I do not see how measurements in Norway can help us constrain
the ice thickness in Greenland or in Antarctica. I mean the setup
is very different. Moreover, there exist a lot of thickness measurements
for both ice sheets. Or do you think of the glaciers outside the
ice sheets?
**L170 :** [...] in [...] $\longrightarrow$ [...] for [...]
**L686-690 :** Please confirm if the Data Availability Section is
part of the main manuscript at ESSD. If not, present this section
together with the acknowledgements, author contributions, etc.
**L692 :** Add a comma after 'In this paper'

FIGURES
**Fig. 1 :** What do the red dots indicate. I did neither find
them in the legend nor in the caption.
**Fig. 1 & Fig 6 :** Consider moving them to the Appendix or a Supplement.
You could directly use Fig. 9 as an overview showing the thickness
measurements. All the other information on radar frequency and
survey type (helicopter, snowmobile, foot) seem less relevant.
**Fig. 3 - Fig. 5 :** Think about only keeping Fig. 3 in the main
manuscript. As much as I appreciate these additional figures, they
could well be suited for an appendix/supplement - possibly by also
transferring associated text blocks/paragraphs.
**Fig. 8 :** I would first present the thickness map (Fig. 10)
and afterwards the uncertainty maps.
**Fig. A1 :** If possible, please add the locations of the 1986/1987
borehole measurements as well as the 1988 GPR surveys.

TABLES

**Table 2** I could not find that the asterisk * information was referred to in the table.  I suspect the last row.

---

## Author Comment (AC1)

**Response letter to ESSD:**
**'Ice thickness and bed topography of Jostedalsbreen ice cap, Norway'**

*by Mette K. Gillespie, Liss M. Andreassen, Matthias Huss and co-authors.*

We thank both reviewers for their constructive comments on our manuscript. We have considered all comments and suggestions for improvements, and here provide our responses as well as a description of the actions we have taken to incorporate them into the manuscript.

Most importantly we have applied the following changes to the manuscript in response to the comments provided by the two reviewers:

1. The description of the modelling approach has been elaborated and clarified, and we now include more information on the limitations and uncertainties associated with the modelling results.

2. To better ascertain the uncertainties in modelling results for regions without measurements, we have run an additional modelling experiment where ice thicknesses along ice divides have been excluded from the input dataset. We include the results of this experiment in the revised manuscript.

3. While ESSD guidelines on data section articles prevent us from including a full comparison of our results with previous measurements and models of ice thickness distribution at Jostedalsbreen ice cap, we now include three new figures and a short discussion to better demonstrate the value of the new ice thickness measurements and glacier model.

4. A figure has been moved from the main article to the appendix and additional details on earlier ice thickness measurements at Austdalsbreen have been added to an existing figure.

In the text below, the original reviewer comments will appear in black font while our responses are in blue font. Quoted sentences and paragraphs from the original submitted manuscript appear in *blue italic font* with our suggested revised text in ***bold blue italic font***, both with a smaller font size.

**Author response to Reviewer 1 ESSD-2024-167**

**General**

Gillespie et al. present an extensive collection of ice thickness observations on Jostedalsbreen ice cap obtained using ground penetrating radar (by scooter, by helicopter and by foot) between 2018 and 2023. They derive an impressive coverage of the ice cap with 90% of the area now being within 300 m of a data point. The point observations are used to calculate a distributed map of ice thickness and bed topography based on a thickness inversion approach. Uncertainties are presented for both the observations as well as the modelled thicknesses. The final ice volume of 70±10.2 km2 is closely aligned with previously published estimates. The distributed bed and thickness maps are thought to form a valuable input for future modelling studies of the ice cap as well as for predicting landscape change, e.g. with regard to future lakes and possible break-up points of the ice cap.

I would like to commend the authors for their efforts to obtain such a wealth and density of thickness observations which may well be unique for an ice cap of this size. A transparent account of error sources and related uncertainties is evident throughout the manuscript and lends trust to presented results. The manuscript is well written and clear, with appropriate figures and tables. The provided data is easily accessible and well documented. All in all, the significant improvements in coverage and accuracy compared to the previously available thickness observations of Jostedalsbreen surely warrant publication in ESSD. I have only minor comments with the exception of one related to the inverse modelling approach.

**Response**: We thank the reviewer for the kind comments on our manuscript.

**2 Specific comments**

**2.1 Inverse modelling**

The inverse approach chosen is based on Huss and Farinotti (2012), a method which reduces glacier shape to a flow-line and is often thought to not be the ideal tool for ice caps (e.g. its comparatively low performance there is already discussed in paragraph 25 of the original publication, but also in, e.g., Millan et al. (2022); Frank and Pelt (2024)). Previously published ice thickness maps of ice caps based on this method (e.g. Farinotti et al., 2019) are not seldom characterized by unrealistic bed shapes and issues at ice divides. Regarding the latter, the authors here aim to alleviate the problems by resorting to an iterative procedure that involves interpolation of nearby GPR profiles and estimated thicknesses on ice divides based on local knowledge" - doesn´t sound entirely convincing.

**Response**: The point brought up by the reviewer is valid and we agree that it requires a more dedicated discussion in the revised paper (see below for detailed actions). However, we would like to raise two important points that may not have been stated clearly enough in the submitted manuscript and therefore seem to have been overlooked by the reviewer:

(1) We do not use the Huss and Farinotti (2012) approach "as is" in our study, but - as described in the Methods section - a further development tailored to optimally assimilating spatially variable sets of point measurements. This approach was first published by Grab et al. (2021), and was also part of the Ice Thickness Models Intercomparison eXperiment phase 2 (ITMIX2; Farinotti et al., 2021). In that paper, various approaches for constraining a distributed ice thickness model with different densities of local thickness data, exactly the problem of the present work, were confronted. The set of models also contained approaches that are more strongly rooted in ice-flow dynamics (e.g. Fürst et al., 2017; Morlighem et al., 2017). The results of ITMIX2 showed that the approach applied in our present work is on a par with these approaches considering all statistics. There is thus no reason to disregard the approach for which we opted here. Of course, we fully agree with the reviewer that applying multiple approaches (based on different physical principles) would yield an optimal result (as also shown by Farinotti et al., 2021). However, the main scope of our data paper in ESSD is to make available a comprehensive data set and not to perform a model intercomparison study. Our dataset of ice thickness point values is available online along with our paper and allows applying any alternative approach to infer ice thickness distribution and bedrock geometry.

(2) Regarding our approach for ensuring ice thickness consistency at ice divides the reviewer seems to have missed one part of our approach. It is described on lines 491-496 of the submitted paper. The addition of "estimated thicknesses on ice divides based on local knowledge" is only relevant for very few cases where the available set of measurements too poorly constrains the result.

In response to the reviewer comment we have revised and clarified the presentation of our approach, improved the description of the procedure at ice divides and added a discussion on the use of alternative approaches for inferring ice thickness distribution.

**Revised text** with context (Introduction): *"For regions that remain unmeasured due to resource or accessibility constraints, we use **inter- and extrapolation of the direct measurements in connection with locally constrained** ice thickness modelling to provide new grids of ice thickness and bed topography for the entire ice cap."*

**Revised text** with context (Methods): *"Here, we apply an approach that combines the advantages of inter- and extrapolation of **point** ice thickness observations with those of ice thickness modelling from an inversion of surface topography (Huss and Farinotti, 2014; Grab et al., 2021). The basis of this approach is an ice thickness model originally developed for global-scale applications (Huss and Farinotti, 2012). The model was used in the Ice Thickness Model Intercomparison eXperiment (ITMIX and ITMIX2; Farinotti et al., 2017, 2021) and performed well in estimations of ice thickness distribution and bed topography **in comparison to a wide range of other approaches. This was the case both if no nearby ice thickness measurements were available, and when such observations were integrated for constraining model parameters."***

*"The general concept of the model **for glaciers without measurements** is to derive local ice thickness from surface characteristics. It relies on glacier surface hypsometry of all individual glacier units of Jostedalsbreen, discretised into 10 m elevation bands. Variations in the valley shape and the basal shear stress along each outlet glacier's longitudinal profile, as well as an estimated **constant basal sliding fraction of 0.5** (e.g., Huss and Farinotti, 2012), are taken into account. Ice volume fluxes are computed along a longitudinal profile based on calibrated mass balance gradients. Subsequently, ice*

*thickness is calculated by inverting the flow law for ice (Glen, 1955), **thus assuming parallel flow consistent with the shallow-ice approximation**. Resulting averages of elevation-band ice thickness are then **interpolated** to a regular grid by considering both local surface slope and distance from the glacier margin, excluding ice divides (for details see Huss and Farinotti, 2012). **For glacier units with ice thickness measurements (i.e., the vast majority of Jostedalsbreen) the model result is first optimised to fit the observations and then only used in unmeasured regions along with all measured point ice thicknesses in an inverse-distance interpolation scheme (see details below).***"

*"**Our approach provides a spatially complete ice thickness and bedrock grid that, per definition, agrees with all thickness observations. We decided to use this methodology rather than approaches based on assimilating the ice flux divergence (e.g. Fürst et al., 2017; Morlighem et al., 2017), as we attribute the highest weight to fitting the comprehensive set of measurements that are at the core of the present study.**"*

**Revised text** with context (Methods): *"... and (3) the condition of zero ice thickness on the glacier margin, except for ice divides. **The total of these point thicknesses delivers a data set that we directly interpolated using an inverse-distance weighting scheme to achieve a full coverage at a 10 m grid spacing.***

*The ice thickness at ice divides was obtained from **interpolated results for neighbouring glacier units**, and also entered the interpolation. **Estimates for ice thickness at ice divides is, thus, given by nearby direct measurements or model results.** Furthermore, **for a few situations with poorly constrained ice divide thicknesses,** a set of individually estimated **point** thicknesses was included to increase the robustness of spatially complete ice thickness and bedrock grid. **These estimated point ice thicknesses were acquired from a direct interpolation of nearby GPR profiles in ArcGIS pro, that involved (1) a 20 m grid spline interpolation (8 sector search radius) of ice thickness measurements and subsequent extraction of 10 m ice thickness contour lines, (2) smoothing of contour lines (50 m smoothing tolerance), and (3) a Topo to Raster interpolation from smoothed contour lines.** Repeating the complete procedure several times ensured convergence and thus consistency of thicknesses on both sides of the ice divides, **thus avoiding thickness steps at ice divides even though glacier units were treated separately in our approach.**"*

Nevertheless, this strategy is helpful when looking at the output, yet there remain small inconsistencies at glacier boundaries. Regarding the overall appearance of the bed shape and thickness distribution, after plotting in QGIS, I find that it does not look realistic. For sure, the thickness observations are well matched, but overall clear "stripes" are visible that appear as if someone had drawn with a thick pencil. So, my general question is: why did the authors settle for this approach and not for other methods on distributed grids, possibly even one that is specifically designed for assimilating thickness observations (e.g. Morlighem et al., 2017; Fürst et al., 2017; Jouvet, 2023)? Looking at the modelled thickness field, I have doubts that the flux divergence would look somewhat realistic when put into a distributed ice flow model. This could turn out to be an issue if, as suggested in the introduction, this bed shape is used for prognostic simulations. In fact, is the final thickness distribution mass conserving in any way? For the plausibility of the location of future lakes, the unrealistic bed shape also doesn´t help (although I am aware that there is always a relatively large uncertainty for such a product).

**Response**: We are well aware that extrapolating a set of measurement points with large variations in spatial density to a regular grid is challenging and that not all locations across the whole ice cap will look entirely plausible. We consider our approach as an optimal

balance between sticking to the ice thickness measurements (the core dataset of this publication) and modelled ice thickness in unmeasured regions. We agree that alternative modelling approaches might come with advantages, but local artefacts or other problems can very likely be detected for every approach when scrutinising the results.

In the revised version we have added more discussion on potential problems and limitations of the ice thickness and bedrock dataset to address this comment by the reviewer. We have already presented some of these revisions (see page 4 in this document). In addition, we add:

**Revised text** (Methods and data): "*We note that beyond the uncertainties estimated above, our data set of gridded thickness and bedrock for entire Jostedalsbreen comes with some limitations that should be considered regarding its usage: We intentionally rely on a statistical inter- and extrapolation of measured point thickness here and supplement this data with results from modelling constrained with the observations in unmeasured regions. This might result in inconsistencies with the application of a three-dimensional ice flow model. Nevertheless, we argue that in the frame of the present publication, whose main emphasis is on measured ice thickness, we strive to optimally make use of these observations and to attribute them with the highest weight in our gridded data set. This also drives the decision to post our results on a 10 m grid, which may imply an exaggerated accuracy for regions without direct measurements but allows resampling to coarser resolutions, depending on the specific application.*"

Besides this general comment on the choice of inverse method, I find that section 3.7 in parts is unclear and should be revised (c.f. specific comments). Simply, while I understand that the method has already been presented in other publications and hence some omission of details is justified, I currently don´t find myself able to understand all steps taken. Not the least, more information on key model parameters should be presented.

**Response**: We have now clarified the description of our applied approach in the revised manuscript and provide more information (see details) under the Specific comments.

**2.2 Other specific comments**

L36f: Slightly re-formulate to make clear that future studies can use the outputs from this work for climate change impact studies. This is not done here.

**Response**: We agree, and the sentence has now been improved.

**Revised text** with context (Abstract): "*These datasets will **be of particular value to future climate change impact studies** in the Jostedalsbreen region, which are of high importance to local stakeholders such as farmers, tourist operators and hydropower companies.*"

L40: There are also other reasons for glacier mass loss than atmospheric temperature increase (e.g. increased ocean temperature for marine-terminating glaciers). Please re-phrase.

**Response**: This has now been clarified in the text.

**Revised text** with context (Introduction): *"Global glacier mass loss caused by increased atmospheric temperatures **and associated processes** contributes significantly to changes in sea level, water resources and natural hazards (IPCC, 2021)."*

L 52: A newer version of the GlaThiDa with continuous additions of published thickness observations is available under https://gitlab.com/wgms/glathida. Consider referring to updated numbers from there as well. More generally, besides publishing the data on the repository given in the manuscript, would you consider actually feeding it into the GlaThiDa?

**Response**: Indeed, we would like our data to be included in GlaThiDa and have prepared the dataset accordingly. The GlaThiDa dataset is however so far released at irregular time intervals. As our dataset is openly available with a DOI, it can be ingested and included in the next release of the GlaThiDa.

L 169f: Here and in other places, the helicopter radar system is described as new/newly developed. However, not much details regarding its novelty are given. So, what exactly are the novelties? And has the system been thoroughly tested beyond what is mentioned in L 365ff? Also, what was the travel speed?

**Response**:  We agree that this may confuse the reader and have rephrased our description of the airborne system. The Air-IPR has been tested by the manufacturer, Blue System Integration Ltd. However, the results of these tests have  not been published so far. In response to the comments, we have added more information on the radar measurements setup.

**Revised text** with context (Methods and data): *"... but a **new** helicopter radar system (Air-IPR) **based on the ground-based Blue System Integration Ltd. IceRadar (Mingo and Flowers, 2010)** was deployed …"*

**Revised text** with context (Methods and data): *"To ensure **a ~30 m** distance between the antennas and the ice surface, we used a laser mounted on the platform with a wireless connection to the cockpit. **Travel speed during the helicopter measurements was ~10 m s$^{-1}$ and** the control of the IceRadar during both ground-based and airborne measurements was performed using a tablet and a remote connection."*

L 274: You mention that you could infer information on glacier thermal regime from the collected GPR data. Was this done systematically, and if so, would it be worthwhile adding more information on that in the manuscript? Could be quite interesting!

**Response**: We have not investigated changes in thermal regime across Jostedalsbreen in more detail than what is described in the manuscript, but we agree that this would be very interesting to explore further in the future! There are some limitations in the majority of the radar measurements because of the low frequency signal (2.5 and 5 MHz antennas), but measurements conducted with 25-500 MHz antennas could provide interesting information on internal glacier characteristics.

L 434: The inversion method was originally developed for global applications and was relying on some large-scale input products and coarse assumptions to infer parameters such as basal sliding, ice viscosity and apparent mass balance. Was this global setup used to infer these parameters here as well, or were they based on more local knowledge/input products (e.g. the mass balance observations on Jostedalsbreen)? Also, in Huss and Farinotti (2012), one basal parameter per glacier was chosen, not a longitudinal trend as stated here. Please state or provide a reference on how the basal sliding distribution was derived.

**Response**: No, this approach was not used here. The concept of applying ice thickness modelling for unmeasured regions of the glacier and tying the model parameters to local measurements or using observations directly when the coverage is dense was clearly described in the submitted version of the paper. We have gone through the description of the methodology again to further clarify this in the text in section 3.6. We have also added more details regarding the assumptions for basal sliding.

**Revised text** with context (Methods and data): "*... as well as an estimated **constant** basal sliding fraction of 0.5*."

L 444: Please state that this integrated form of Glen's flow law assumes parallel flow and thus in essence is the shallow ice approximation. In my view, this information is important for readers to be able to evaluate potential errors of the model output.

**Response**: Done

**Revised text** with context (Methods and data): "*Subsequently, ice thickness is calculated by inverting the flow law for ice (Glen, 1955), **thus assuming parallel flow consistent with the shallow-ice approximation**.*"

L 449: Could you please comment on your choice of inferring ice thickness on a 10 m grid? There is a physical limit to how much spatial detail one can obtain from a thickness inversion, at best horizontal features of >1 times the ice thickness can be resolved (e.g. Gudmundsson, 2003). Isn´t the choice of a 10 m grid unnecessary? Or even, doesn´t it give the wrong impression that such small features should actually be detectable in your thickness map? You later mention that you smooth the bed topography (but not the thicknesses, why?) with a 50-100 m filter which, however, still in many places is lower than one ice thickness.

**Response**: This is a valid comment. The choice of extrapolating ice thickness to a 10 m grid was driven by the availability of the surface digital elevation model that is provided at 10 m resolution. To work with a DEM at this posting, it was obvious to extract thickness distribution at the same resolution. Indeed, this might not be the actual information content of the grid, even though regions directly observed with GPR lines feature such a degree of detail in 1-D. To facilitate working with the data set in all possible applications, we would still like to maintain the 10 m resolution that allows resampling to any coarser resolution. As suggested by the reviewer's comments, we now discuss limitations of this approach (see below).

To respond more specifically to the reviewer's comment: As now even more clearly emphasised in the text, the thickness grid presented here is not a thickness inversion (in the sense of Gudmundsson, 2003), but in almost all regions of the ice cap directly given, or at least closely constrained by measurements (see Fig. 9). The text explains why we smooth the bedrock topography but not the ice thickness ("… *to remove remaining discontinuities at ice divides, as well as unrealistic small-scale variability in calculated bed topography that cannot be inferred with the applied methodology and will originate from surface features.*"). Of course, ice thickness grids are afterwards updated to be consistent with surface topography and inferred bedrock.

In response to the comment, we have now added a discussion of this good point raised by the reviewer, in connection with other, similar issues raised in the review (see page 5 in this document for the revised text).

L 463ff: Here you mention that the apparent mass balance gradient is optimized to reduce the misfit to the thickness observations. In the next sentence, there is talk of two gradients that are subsequently computed. Please clarify the discrepancy on the number of gradients. Furthermore, it is not clear to me why the apparent mass balance is first optimized, and then calculated again. What exactly is done when it is calculated after the optimization? Lastly, you assume a balanced mass budget over the entire glacier unit. Instead of an assumption, isn´t that a necessity that follows from mass conservation?

**Response**: The paragraph is reformulated and clarified in the revised document.

**Revised text** with context (Methods): *"In step (i), we optimised the apparent mass-balance gradient (Farinotti et al., 2009) for the ablation and accumulation area, assuming a constant ratio of 1.8 between the gradients, in an automatic procedure to minimise the average misfit between modelled ice thickness and the available observations for each of the 59 outlet glaciers with ice thickness measurements. To close the mass budget, we prescribed a balanced mass budget for the entire glacier unit (see Farinotti et al., 2009). The resulting apparent mass balance distribution was then used to compute ice volume fluxes from the top to the bottom of each glacier unit, and to infer modelled ice thickness distribution."*

L 473ff: Is the difference field smooth? It sounds as if that wouldn´t necessarily be the case, e.g. one observation point may be well matched, whereas the neighboring one isn´t. How does that affect the extrapolation? And why is the ice thickness distribution after this step not going through the observations, if you apply point-wise corrections? And on a different note: I am somewhat confused about the terms extra- and interpolation (here and in other places of this section). Please clarify what you mean when you interpolate vs when you extrapolate, and between what or where to you do that.

**Response**: The difference field must be smooth as medium- to large-scale information on misfits are extracted from the model-measurement comparison. We would like to thank the reviewer for the notion of the inconsistent handling of inter- and extrapolation. We homogenised this in the revised version, even though both terms remain to be used to differentiate interpolation between measured points and extrapolation to unmeasured regions. The description is reformulated and clarified.

**Revised text** with context (Methods): *"First, relative differences between modelled and measured point ice thickness distributions were evaluated. These differences were then spatially inter- and extrapolated based on an inverse-distance weighting scheme **that results in a smooth field over the entire glacier and allows extracting large-scale spatial variations in misfits**. This relative spatial ice thickness correction field was then superimposed on the modelled ice thickness distribution, resulting in a bias-corrected model-based ice thickness distribution that accounts for the differences between observed and modelled ice thickness at a spatially distributed scale."*

L 479ff: Here I cannot follow any longer. I take that after step ii, you already have a thickness distribution that is based on all available thickness observations. However, now again thickness observations in combination with model results are used for some more interpolation. Why and how is that done? Where do you interpolate between? What happens within the buffer? Do you simply set the thickness there to the observed one? And ultimately, is your final thickness distribution consistent with the ice dynamics and apparent mass balance of the inversion? If not, what distinguishes your approach from a statistical interpolation?

**Response**: The description is reformulated and clarified. The "How" is now better described. The "Why" is explained when introducing the method: We would like to have our gridded thickness product in full agreement with the measurements, which is the core of the present study. The locally optimised model delivers thickness estimates for all unmeasured regions - our approach combines these two data sources in an optimal way.

The last question by the reviewer is interesting. In regions that are well covered with measurements (a large fraction of the ice cap in this study), our approach indeed is a simple statistical extrapolation based on inverse-distance weighting of the GPR measurement points. Only in regions that are not covered (within a spatially variable buffer) with direct observations, (optimised) ice thickness model results are used. We therefore deliberately give a higher weight to the agreement of the ice thickness grid with local observations, than to preserving flux divergence from an ice dynamical point of view.

**Revised text** with context (Methods): *"In the final step (iii), we spatially interpolated the ice thickness distribution based on (1) all available ice thickness observations, (2) the model results adjusted in steps (i) and (ii) in regions that were not covered by direct measurements (buffered in a distance of 100–200 m around available observations depending on outlet glacier size), and (3) the condition of zero ice thickness on the glacier margin, except for ice divides. **The total of these point thicknesses delivers a data set that we directly interpolated using an inverse-distance weighting scheme to achieve a full coverage at a 10 m grid spacing.**"*

Further revisions related to these issues have been addressed earlier (see page 5 in this document for the revised text).

L 484: Please provide more information on what is meant by "local knowledge" and how that information was transformed to quantitative values.

**Response**: The updated manuscript now includes a more detailed description of the interpolation of point ice thicknesses along ice divides as described earlier in the response (see page 4 in this document for the revised text).

L 516: Another way of assessing uncertainties would be to remove some of the thickness observations from the data set, re-run the inversion and test how the resulting thickness field compares at those spots. Could that be worthwhile?

**Response**: Yes, this is a good suggestion. This experiment has now been conducted and the difference in the final result has been analysed, with an additional statement in the revised paper.

**Revised text** (Methods): *"To assess the relevance of additionally defined thickness points along ice divides used to better constrain the thickness inter- and extrapolation in these regions (see Section 3.6.), we performed an experiment where these supporting points were removed. We find that the effect on the inferred total ice volume of Jostedalsbreen is minimal (-1.1%), and local thicknesses are affected by 1.2 m on average (median absolute difference)."*

L 594f: You mention that the old data set has considerable uncertainties in many places, and thus you limit your comparison to an area which is more certain. I would argue that there is no reason for that. Indeed, to get a full understanding of how much better your new data set is plus to allow users of the old data set to judge how "bad" the old data was, I think it would be meaningful to undertake a full comparison. A simple scatter plot of old vs new thicknesses (at places where the two are reasonably close together or using your interpolated bed map) would already be helpful in my view.

**Response**:  Unfortunately, we do not have all the older measurements available as point data so a direct intercomparison is not possible in all regions. However, to better illustrate the differences between the previous and new datasets in spatial coverage and measured ice thickness, the revised manuscript includes a new figure that compares measurements in the southern part of Jostedalsbreen. The figure includes previous point measurements from 1984 and 1989, the latter of which is included in GlaThiDa (2020). We have also added a short description of the differences and similarities observed in the figure.

**Revised text** with context (Results): *"The new comprehensive dataset of Jostedalsbreen ice thicknesses represents a significant improvement to previous measurements, both in relation to data quality and spatial coverage across the ice cap. We now have a much better understanding of ice thickness variations in the region and have also extended the maximum measured ice thickness from 600 m measured during the 1980s field campaigns (Sætrang and Wold, 1986) to the 631 m measured in 2021.* **Although the general ice thickness variability identified in the new measurements are also recognisable in the older datasets, distinct differences are observed across the ice cap (Fig. 9). Regions with thick ice are particularly poorly resolved in the earlier measurements, most likely due to limitations in the radar system applied during these field campaigns. While we believe that most of the discrepancies can be attributed to measurement uncertainties, evidence of glacier retreat since the measurements in 1989 is discernible in marginal regions.***"

[Figure]

**Revised text** (new Figure 9 caption): *"(a) Previous ice thickness measurements collected in the southern part of Jostedalsbreen in 1984 and 1989. Only the 1989 dataset is included in GlaThiDa (GlaThiDa consortium, 2020) due to large positioning uncertainties in the 1984 measurements. (b) Ice thickness measurements collected during the 2018, 2021, 2022 and 2023 field seasons. Locations of maximum measured ice thickness during the respective field campaigns are marked on both figures. The 1966 outline of Jostedalsbreen is from Paul et al. (2011) and the 2019 outline is from Andreassen et al. (2022). The coordinate system on both figures is UTM 33N, datum ETRS_1989."*

**3 Technical comments**

L 26: I think adding a reference for the GlaThiDa is appropriate here, as not all readers can be assumed to know what that is.

**Response**: This has now been added.

L 62: Fürst et al. (2017) actually assimilates thickness observations, so should not be mentioned here

**Response**: Thanks, the reference is removed.

L 74: remain, not remains

**Response**: Thank you. This has been corrected.

L 141: "continues to this day": consider rephrasing, e.g. "continue to do so to this day"

**Response**: We agree and have rephrased accordingly.

L149: In contrast to other place/glacier names mentioned in the text, Bødalsbreen is not labelled on the figures

**Response**: Thank you for noticing this. We have now labelled Bødalsbreen on our Figure 1.

L 629: Consider including also a comparison to Farinotti et al. (2019) and Millan et al. (2022)

**Response**: As also suggested by Reviewer 2, a comparison to other, global data sets has now been performed, including a short discussion of the results. This is included as two new figures included later in this document (Fig. 11 and Fig. 12).

**4 Figures and Tables**

Table 2: In the GlaThiDa, the attribute field for elevation is ELEVATION, whereas it is GNSS ELEVATION here. For simplicity of usage, consider aligning with the GlaThiDa

**Response**: We have chosen to label the radar measurements of elevation "GNSS ELEVATION" so as to clearly distinguish this attribute from the surface elevations provided by the 2020 DTM. The latter is used in our calculations of bed topography. We prefer to maintain the attribute titles as is, but we will of course use ELEVATION as the name of this attribute field when submitting the data to GlaThiDa.

**Author response to Reviewer 2 ESSD-2024-167**

Summary and comments on the manuscript essd-2024-167 entitled **Ice thickness and bed topography of Jostedalsbreen ice cap, Norway**

Presented on 06.05.2024 by Mette Gillespie et al.

**SUMMARY**

The authors present an extensive record of ground penetrating radar (GPR) measurements to map the glacier ice thickness of Jostedalsbreen collected during multiple ground and airborne field campaigns between 2018 and 2023. Data was collected with various radar systems involving operation frequencies between 2.5 and 500 MHz. The acquired record is impressive with more than 350'000 point measurements and more than 1'000 line kilometers surveyed. These measurements are ingested into an inverse procedure to infer a high-resolution map of glacier ice thickness for the entire glacierised area. This map is truly valuable, as previous reconstructions of glacier ice thickness had no measurements at their disposal. Moreover, the authors provide an uncertainty map of their glacier ice-thickness field. A brief analysis of the basal topography for sub-glacial over-deepenings - i.e., possible sites for future lake formation - completes the manuscript.

I was very excited about this article and I want to admit that I was at no point disappointed. I want to congratulate the authors to this piece of work. The manuscript is very well written and strikes with clearness and high-quality illustrations. Below you find some major comments on several aspects of the methods and analysis. None of them are fundamental but will help to improve or to better assess your results. Overall, I am very positive about this manuscript because it combines '*impressively extensive*' field observations with modelling techniques. I therefore recommend that the editor should continue to considered this manuscript for publication in *Earth System Science Data* after minor revisions.

**Response**: We thank the reviewer for the kind comments on our manuscript.

**MAJOR COMMENTS**

OPTIMAL RECONSTRUCTION

You use an approach by Huss and Farinotti (2012) (HF apporach) to infer the glacier ice thickness of Jostedahlsbreen. Several parameters in this approach are not well constrained. How did you select the optimal parameters with regard to ice-dynamics as well as mass overturning/surface mass balance. Couldn't you use your abundant measurement record for a dedicated calibration.

**Response**: As for reviewer 1, we have the impression that some basics of the approach applied was misunderstood, and we would like to apologise that our description seemingly was not clear enough. Even though the original Huss and Farinotti (2012) is at the basis of the methodology, a completely different implementation is used here: The abundant ice thickness observations are used - as suggested by the reviewer - to optimally constrain all parameters of the model. Furthermore, the model results are only considered in regions that are not covered by direct ice thickness measurements. Our result is thus a fusion of an optimally constrained model and the measurements. We emphasise this important aspect in the revised manuscript following comments from both reviewers. See page 3 – 4 in this document for our edits to the manuscript.

THICKNESS HOMOGENISATION

As you have a second DEM from 2017 (aside the 2020 DEM) you could infer an elevation change map (possible a co-registration is necessary). So you could refrain from using individual elevation differences from DGNSS measurements to homogenise your thickness data. In this way, you account for spatial difference in elevation change. An elevation change map would further be useful for my comment on the 'future assessment' (below).

**Response**: The 2017 elevation map is not considered to be very accurate by the Norwegian mapping authorities, which is why the glacier was surveyed in 2020 using airborne laser scanning. Furthermore, we do not think this approach would help in our context: There are no radar measurements from 2017, but many acquired radar measurements come with

accurate DGNSS measurements of the elevation that can directly be used to homogenise simultaneously observed thicknesses to the 2020 DEM.

FUTURE ASSESSMENT

I think for the potential disconnection of Jostedalsbreen (L574-587), you have to combine your thickness observations with actual elevation changes observed by satellite remote sensing. I say that because even thin ice can prevail for a long time at high elevation. Moreover, I would rather use the thickness map to analyse future disconnection possibilities - instead of the point measurements.

**Response**: We deliberately focus mainly on the measured results in our data paper, but we have now added a reference to the observed glacier changes in this section.

**Revised text** with context (Methods): *"In the north, Lodalsbreen currently connects the northernmost part of Jostedalsbreen with its more southern regions through three steep tributaries (Fig. 9b). Helicopter measurements along the centre flowlines reveal that the ice thins to 50 m or less as it flows southwards and into the incised valley below. Ice flowing from the western tributary is thicker, with ice thicknesses ranging between 50 and 70 m along its thinnest sections. **A study of surface elevation changes at Jostedalsbreen between 1966 and 2020 shows that the glacier has experienced significant thinning in this region (Andreassen et al., 2023). This trend is likely to continue as Jostedalsbreen adjusts to warmer air temperatures.**"*

UNCERTAINTY MAP

I remain confused about how you built up this final map of thickness uncertainty (Sect. 3.8). You first produce additional thickness fields by feeding the minimum and maximum thickness estimates from your observations (relying on the measurements error) into the HF approach. For the extrapolation uncertainty, you vary certain model parameters. Yet it is unclear how many parameter combinations you tried and how you sample. You stay rather vague here. Moreover, I did not find which measure you used to quantify the variability (min/max, sigma, ...). Lastly, it is not clear to me how you combined the measurement error maps with these extrapolation uncertainty maps to produce a final uncertainty map. Please be more specific.

**Response**: We agree with the reviewer that this description needs refining. Therefore, we have implemented the corresponding changes in the revised paper.

**Revised text** with context (Methods): *"The uncertainty in **inter- and** extrapolated ice thickness is composed of two elements: (1) the uncertainty in measured ice thickness, and (2) the uncertainty induced when extrapolating point ice thickness across the entire ice cap supported by the model-based approach. These two elements of uncertainty are estimated **with separate experiments, and are** then propagated through the methodology described above to derive a spatially distributed uncertainty map for the entire ice cap."*

*As described in section 3.4, the uncertainty associated with each point value of ice thickness was calculated following Laparazan et al. (2016). We conservatively assume all uncertainties across the entire ice cap to be correlated and generate a dataset with maximum **and minimum observed ice thickness** according to the above uncertainties. Based on these two datasets, we repeated the **complete** approach described in section 3.6 using each of these datasets. **Two additional experiments were conducted** to assess the uncertainty caused by extrapolating observations to unmeasured regions. **Relevant parameters of the ice thickness model were set to the maximum or the minimum of conservative** but physically meaningful ranges. This was performed for the viscosity of ice, the assumed fraction of basal sliding, and the apparent mass balance gradients. In **both** experiments, the reference dataset of point ice thickness values was used for calibration **(see Section 3.6)**, such that the resulting ice thickness grids differ mostly in regions where thickness is solely inferred by the model.*

*Finally, we combined the offset from the reference ice thickness **at all grid cells for the four experiments described above (two for measurement uncertainty, two for model uncertainty)** based on the root-sum-of-squares. **This results** in an absolute and a relative uncertainty grid. …"*

DISCUSSION

You state that the volume is very similar to previous estimates. I strongly doubt that these previous estimates did rely on as many thickness measurements as you had. So why are these estimates so similar. Do we no longer need to conduct measurements? I strongly doubt that. I think there must be quite some differences in the thickness distribution - worth to discuss. Did these approaches use thickness observations in this region? GlaThiDa 3.1.0 holds no data on Jostedalsbreen. What about a comparison with the global products from Farinott et al. (2019) and Millan et al. (2022), that many people do use. I think that a map comparison of ice thickness is a worthwhile effort here.

**Response**: There are several important aspects in this comment. First, we would like to state that GlaThiDa 3.1.0 indeed holds data on Jostedalsbreen. Data from 1989 are included in the T (row 619) and TTT spreadsheet (row 97129-98197). In sheet T data is described as:

'*Andreassen et al., (2015). DOI: 10.3189/2015JoG14J161",,"GLACIER_ID references the Norwegian Water Resources and Energy Directorate (NVE) atlas. Surveys 15-17 April 1989. A thinned data set. Measurements include winter snow. Point data of southern part of Jostedalsbreen, including Tunsbergdalsbreen, Austerdalsbreen and Bøyabreen. …'*

```
97129  2108,NO,JOSTEDALSBREEN,19890416,,1,61.6763177,7.0291111,1902,61,,,
97130  2108,NO,JOSTEDALSBREEN,19890416,,2,61.6753207,7.0241101,1883,161,,,
97131  2108,NO,JOSTEDALSBREEN,19890416,,3,61.6713584,7.0047688,1850,469,,,
97132  2108,NO,JOSTEDALSBREEN,19890416,,4,61.6698032,6.9972736,1837,330,,,
97133  2108,NO,JOSTEDALSBREEN,19890416,,5,61.6677326,6.9877343,1807,339,,,
97134  2108,NO,JOSTEDALSBREEN,19890416,,6,61.6630593,6.9644063,1784,207,,,
97135  2108,NO,JOSTEDALSBREEN,19890416,,7,61.6623633,6.9610897,1778,143,,,
97136  2108,NO,JOSTEDALSBREEN,19890416,,8,61.660762,6.9536773,1732,175,,,
97137  2108,NO,JOSTEDALSBREEN,19890416,,9,61.6584222,6.9419705,1696,173,,,
```

The statement that the new estimate for the ice volume of Jostedalsbreen ice cap is similar to the previous numbers by no means implies that the many new radar acquisitions are in vain. It is known from previous research that estimating the total volume of a glacier or ice cap is often relatively robust and can also be achieved with less measurements, or simplified approaches. However, with the present data, for the first time, a complete coverage of data with a high confidence has been compiled, uncertainties have been reduced and the spatial distribution of the ice thickness is now much better constrained than before. We included a short version of this discussion in the paper.

We agree that the suggested comparison to global-scale results by Farinotti et al (2019) and Millan et al (2022) is important to our study. We have now included two figures in the manuscript (Fig. 11 and 12) as well as a short discussion for comparing our results to these data sets. However, we emphasise that the main aim of this data paper in ESSD is to present the data set of GPR measurements and to make these observations available to the scientific community for further studies.

[Figure]

**Revised text** (new Figure 11 caption): *"Comparison of measured and modelled point ice thickness across Jostedalsbreen according to the large-scale ice thickness model data sets by (a) Farinotti et al. (2019), (b) Millan et al. (2022), and (c) Frank and van Pelt (2024). Comparisons are limited to locations within the respective model grid and calculated mean error (in metres) is negative when modelled ice thicknesses exceed measured ice thicknesses. The black line in each figure indicates the 1:1 line."*

[Figure]

(a) Farinotti et al. (2019)

(b) Millan et al. (2022)

(c) Frank and van Pelt (2024)

(d) This study

0 — 100
101 — 200
201 — 300
301 — 400
401 — 500
501 — 600
601 — 700

0    10 km

N

**Revised text** (new Figure 12 caption): *"Ice thickness distribution on Jostedalsbreen according to the large-scale model studies by (a) Farinotti et al. (2019), (b) Millan et al. (2022), (c) Frank and van Pelt (2024), and (d) this study."*

**Revised text** with context (Results): *"Overall, the presented results are consistent with previous estimates for* **the volume and the thickness distribution** *of Jostedalsbreen. The calculated mean ice thickness is slightly smaller than the earlier estimate of 158 m which was calculated for an interpolated region covering 65 % (310 km$^2$) of the 2006 area (474 km$^2$) of Jostedalsbreen (Andreassen et al., 2015). Our calculated ice volume* **(70.6 km$^3$)** *also compares well with* **previous volume estimates of 69.6 km$^3$ and 68.5 km$^3$ provided by Farinotti et al. (2019) and Frank and van Pelt (2024) respectively, while the ice thickness model proposed by Millan et al. (2022) appears to underestimate the ice thickness at Jostedalsbreen, with a calculated volume of 56.5 km$^3$. A comparison of our point thickness measurements with modelled values from the respective studies, indicates a standard deviation of between 75 and 90 m, while the mean error is small for Farinotti et al. (2019), and indicates too small and somewhat too high ice thicknesses for Millan et al. (2022) and Frank and van Pelt (2024), respectively (Fig. 11). The ice thickness distribution shows that all large-scale ice thickness models capture the general pattern. However, the results of Farinotti et al. (2019) reveal unrealistic values along the ice divides (Fig. 12a). The inferred thicknesses by Frank and van Pelt (2024) shows a tendency to overestimate thickness on outlet glacier tongues, while the result by Millan et al. (2022) underestimates thickness both in glacial troughs and in the interior of the ice cap. Our comprehensive data set of thickness measurements is expected to improve regional to global-scale assessment of ice thickness distribution by supporting the calibration and validation of ice thickness models."*

STRUCTURE

In the uncertainty subsection of the extrapolated map product (3.8), you present already quite some results. Please transfer these to Section 4.

**Response**: We agree and have now moved the maps of uncertainty and associated description to section 4.

**MINOR COMMENTS**

L757 : I do not see how measurements in Norway can help us constrain the ice thickness in Greenland or in Antarctica. I mean the setup is very different. Moreover, there exist a lot of thickness measurements for both ice sheets. Or do you think of the glaciers outside the ice sheets?

**Response**: We have now removed this sentence.

L170 : [...] in [...] −→ [...] for [...]

**Response**: Done

L686-690 : Please confirm if the Data Availability Section is part of the main manuscript at ESSD. If not, present this section together with the acknowledgements, author contributions, etc.

**Response**: The Data Availability section will be part of the main manuscript, which is structured according to the ESSD manuscript composition guideline.

L692 : Add a comma after 'In this paper'

**Response**: Done - thank you.

FIGURES

Fig. 1 : What do the red dots indicate. I did neither find them in the legend nor in the caption.

**Response**: Thank you for noticing this. We have now added information on the red dots in the figure caption and added a missing reference to Figure 1 at the first mentioning of "Steinmannen".

**Revised text** with context (Figure 1 caption): *"... (b) Jostedalsbreen and GPR surveys divided into helicopter, snowmobile, and foot,* **with red dots indicating locations referenced in the text***, and ..."*

Fig. 1 & Fig 6 : Consider moving them to the Appendix or a Supplement. You could directly use Fig. 9 as an overview showing the thickness measurements. All the other information on radar frequency and survey type (helicopter, snowmobile, foot) seem less relevant.

**Response**: We would prefer to keep Figures 1 and 6 in the main manuscript as we find it easier for the reader and would like to introduce Figure 1 before the results.

Fig. 3 - Fig. 5 : Think about only keeping Fig. 3 in the main manuscript. As much as I appreciate these additional figures, they could well be suited for an appendix/supplement - possibly by also transferring associated text blocks/paragraphs.

**Response**: We have moved Figure 5 to the appendix (now Figure B1), but we have chosen to keep both Figure 3 and 4 in the main manuscript. We find Figure 4 important as it illustrates both the deepest measured point and the limitations in steep terrain even for the 2.5 MHz antenna. The fact that we have a strong reflector at the deepest point adds valuable credit to the dataset.

Fig. 8 : I would first present the thickness map (Fig. 10) and afterwards the uncertainty maps.

**Response**: As described above, we agree and have now moved the maps of uncertainty and associated description to section 4 after the ice thickness and bedrock maps.

Fig. A1 : If possible, please add the locations of the 1986/1987 borehole measurements as well as the 1988 GPR surveys.

**Response**: We have now added the location of earlier radar measurements and boreholes at Austdalsbreen to Figure A1.

[Figure]

**Revised text** with context (Figure A1 caption): "*(a) Locations of ice thickness measurements divided into survey year and (b). ice thickness measurements on Austdalsbreen, including the locations of the 1988 survey lines and boreholes from 1986 and 1987. The coordinate system on both maps is UTM 33N, datum ETRS89. The background imagery in (b) is from Esri (https://services.arcgisonline.com/ArcGIS/rest/services/World_Imagery/MapServer) and in this area relies on a Maxar mosaic with images from 2019 and 2021.*"

TABLES

Table 2 I could not find that the asterisk ∗ information was referred to in the table. I suspect the last row.

**Response**: Thank you for noticing this. You are correct and we have now added the asterisk to the description of the last row.

**References:**

Andreassen, L. M., Nagy, T., Kjøllmoen, B., and Leigh, J. R.: An inventory of Norway's glaciers and ice-marginal lakes from 2018–19 Sentinel-2 data, J. Glaciol., 68, 2022.

Andreassen, L. M., Carrivick, J. L., Elvehøy, H., Kjøllmoen, B., Robson, B. A., and Sjursen, K. H.: Spatio-temporal variability in geometry and geodetic mass balance of Jostedalsbreen ice cap, Norway, Ann. Glaciol., 1–18, 10.1017/aog.2023.70, 2023.

Farinotti, D., Huss, M., Bauder, A., Funk, M., and Truffer, M.: A method to estimate the ice volume and ice-thickness distribution of alpine glaciers, J. Glaciol., 55, 422–430, 2009.

Farinotti, D., Brinkerhoff, D. J., Fürst, J. J., Gantayat, P., Gillet-Chaulet, F., Huss, M., Leclercq, P. W., Maurer, H., Morlighem, M., and Pandit, A.: Results from the ice thickness models intercomparison experiment phase 2 (ITMIX2), Front. Earth Sci., 8, 571923, 2021.

Fürst, J. J., Gillet-Chaulet, F., Benham, T. J., Dowdeswell, J. A., Grabiec, M., Navarro, F., Pettersson, R., Moholdt, G., Nuth, C., and Sass, B.: Application of a two-step approach for mapping ice thickness to various glacier types on Svalbard, Cryosphere, 11, 2003–2032, 2017.

GlaThiDa Consortium: Glacier Thickness Database 3.1.0. World Glacier Monitoring Service, Zurich, Switzerland, 10.5904/wgms-glathida-2020-10, 2020.

Glen, J. W.: The creep of polycrystalline ice, Proceedings of the Royal Society of London. Series A. Mathematical and Physical Sciences, 228, 519–538, 1955.

Grab, M., Mattea, E., Bauder, A., Huss, M., Rabenstein, L., Hodel, E., Linsbauer, A., Langhammer, L., Schmid, L., and Church, G.: Ice thickness distribution of all Swiss glaciers based on extended ground-penetrating radar data and glaciological modeling, J. Glaciol., 67, 1074–1092, 2021.

Gudmundsson, G. H.: Transmission of basal variability to a glacier surface, J. Geophys. Res., 108, 2253, 10.1029/2002JB002107, 2003.

Huss, M. and Farinotti, D.: Distributed ice thickness and volume of all glaciers around the globe, J. Geophys. Res. Earth Surf., 117, 2012.

Huss, M. and Farinotti, D.: A high-resolution bedrock map for the Antarctic Peninsula, Cryosphere, 8, 1261–1273, 2014.

Mingo, L. and Flowers, G. E.: An integrated lightweight ice-penetrating radar system, J. Glaciol., 56, 709–714, 10.3189/002214310793146179, 2010.

Morlighem, M., Williams, C. N., Rignot, E., An, L., Arndt, J. E., Bamber, J. L., Catania, G., Chauché, N., Dowdeswell, J. A., Dorschel, B., Fenty, I., Hogan, K., Howat, I., Hubbard, A., Jakobsson, M., Jordan, T. M., Kjeldsen, K. K., Millan, R., Mayer, L., Mouginot, J., Noël, B. P. Y., O'Cofaigh, C., Palmer, S., Rysgaard, S., Seroussi, H., Siegert, M. J., Slabon, P., Straneo, F., van den Broeke, M. R., Weinrebe, W., Wood, M. and Zinglersen, K. B.: BedMachine v3: Complete bed topography and ocean bathymetry mapping of Greenland from multibeam echo sounding combined with mass conservation, Geophys. Res. Lett., 44, 11,051–11,061. 10.1002/2017GL074954, 2017.

Paul F., Andreassen L.M., and Winsvold S.H.: A new glacier inventory for the Jostedalsbreen region, Norway, from Landsat TM scenes of 2006 and changes since 1966, Ann. Glaciol., 52, 153-162, 10.3189/172756411799096169, 2011.

Sætrang, A. C. and Wold, B.: Results from the radio echo-sounding on parts of the Jostedalsbreen ice cap, Norway, Ann. Glaciol., 8, 156–158, 1986.

---

## Author Response (AR2)

4 November 2024

Dear editor,

We have now uploaded the final files for publication in ESSD. During our final review of the paper, we noticed a few things that we would like to improve if possible. In addition to smaller edits, we suggest making the following changes to the manuscript:

Line 145 and reference list: Updated the Kjøllmoen et al., in prep. with the now published reference.

Line 164: Clarification on the source of data for modelling study by Fran and van Pelt (2024).

Line 327 – 328 and 336: Included a better clarification as to why we describe calculations of glacier bed elevations.

Line 879 – 881: Authors were missing from this reference and have now been added.

Figure 8: Improved resolution of survey lines and deleted the statement on positioning uncertainties in the figure description.

Figure 11: Suggested update to include information on point density. This allows the reader to better understand the differences between the statistics in the model comparison. No additional text is required in the manuscript to describe this figure, but the significance of the colours in the plot should be mentioned as suggested in the figure text if this change is accepted.

Figure A1: Updated legend to include all elements on the figure.

The files that are uploaded in the form includes these changes. If these changes go beyond what you think is reasonable at this stage, then please let us know, and we will submit a version with only the minor edits.

Kind regards,

Mette Gillespie and co-authors